# Balancing water column and sedimentary <sup>234</sup>Th fluxes to quantify coastal marine carbon export

Madeline G. Healey<sup>1</sup>, Erin E. Black<sup>2</sup>, Christopher K. Algar<sup>1</sup>, Maria Armstrong<sup>1</sup>, Stephanie S. Kienast<sup>1</sup>

<sup>1</sup>Department of Oceanography, Dalhousie University, Halifax, Nova Scotia, Canada

<sup>2</sup>Department of Earth and Environment, University of Rochester, Rochester, New York, USA

Correspondence to: Madeline G. Healey (<a href="mailto:mhealey@dal.ca">mhealey@dal.ca</a>)

Abstract. Quantitative estimates of particulate organic carbon (POC) flux and burial in coastal systems are critical for constraining coastal carbon budgets and understanding their role in regional and global carbon cycling. In this study, POC export fluxes were quantified in the Bedford Basin, a coastal inlet in the Northwest Atlantic, based on measurements of  $^{234}$ Th/ $^{238}$ U disequilibria and the POC: $^{234}$ Th ratio on small (1–51 µm) and large (> 51 µm) particles. These water column export fluxes were compared to sediment accumulation fluxes of  $^{234}$ Th and POC to refine the carbon budget in the Bedford Basin. The coupled water column-surficial sediment sampling approach, which is relatively new, was applied quasi-seasonally throughout 2021–2024, including for the first time in boreal winter. Total  $^{234}$ Th activities reveal persistent deficits with respect to  $^{238}$ U throughout the water column, likely due to extensive particle scavenging, which is also indicated by high  $^{234}$ Th activities on particles. Here, we find that the removal of  $^{234}$ Th in the water column is typically balanced, within uncertainties, by the inventory of excess  $^{234}$ Th in underlying marine sediments. This finding reveals that on the timescale of ~100 days, the  $^{234}$ Th budget in the Bedford Basin is in balance, and no major particle loss is occurring. Using the POC: $^{234}$ Th ratio on sinking particles and integrated  $^{234}$ Th water column fluxes, we report a mean ( $\pm$  s.d.) depositional flux of  $20 \pm 14$  mmol C m $^{-2}$  d $^{-1}$  (range 3.6 to 44.5 mmol C m $^{-2}$  d $^{-1}$ ) to the seafloor. This mean flux translates to an annual molar flux of  $7.3 \pm 5.1$  mol C m $^{-2}$  yr $^{-1}$ , within a factor of 2 of model estimates and previous sediment trap results at this site. Our findings contribute to ongoing research efforts in the Bedford Basin, and aid in the evaluation of coastal regions in local carbon budgets.

## 1. Introduction








Coastal and continental shelf zones represent only  $\sim 7$ –10% of the global ocean surface area but host up to 30% of global marine primary production (Muller-Karger et al., 2005) and are important for the long-term burial of organic matter. Coastal areas are dynamic transition zones between land and ocean, where biogeochemical processes modify carbon cycling. In turn, these processes influence air-sea carbon dioxide (CO<sub>2</sub>) exchanges and thus exert a significant control on regional and global carbon cycles. A series of biologically mediated processes referred to as the biological carbon pump (BCP) are important in coregulating atmospheric CO<sub>2</sub> concentrations. In the sunlit surface ocean, primary producers photosynthesize inorganic CO<sub>2</sub> into particulate organic carbon (POC) at a rate of  $\sim 50$  Pg C yr<sup>-1</sup> (Longhurst et al., 1995). A portion of this POC leaves the surface ocean and sinks to depth, whereby a small fraction ends up sequestered in marine sediments. The traditional method of estimating sinking POC fluxes is based on sediment traps, which suffer from conveyance issues and hydrodynamic biases (Butman, 1989). Additionally, living organisms ("swimmers") can enter traps and bias particle flux estimates, an issue which is more severe in the shallow waters characterizing coastal and shelf areas compared to the open ocean (Buesseler et al., 2007). Given these methodological constraints, the confluence of terrestrial and marine carbon sources, and temporal and spatial heterogeneity, the processes controlling carbon cycling in coastal and shelf regions remain elusive.

First proposed by Bhat et al. (1969), the disequilibrium between the naturally occurring radioisotopes thorium-234 (<sup>234</sup>Th, t<sub>1/2</sub> = 24.1 d) and uranium-238 (<sup>238</sup>U, t<sub>1/2</sub> = 4.47 x 10<sup>9</sup> yr) is commonly measured to quantify sinking fluxes of carbon and other elements in the ocean. In shallow regions, the removal of <sup>234</sup>Th from the water column results in an excess of <sup>234</sup>Th (<sup>234</sup>Th<sub>xs</sub>) on the seafloor in marine sediments (Aller and Cochran, 1976), offering a unique opportunity to explore one-dimensional (1D) particle dynamics and export. A recent global oceanic compilation identified 223 studies that applied the <sup>234</sup>Th/<sup>238</sup>U disequilibrium method, encompassing a total of 1981 unique stations (Ceballos-Romero et al., 2022). However, in nearshore coastal settings, defined here as within 5 nautical miles (nm) of the coast, there are comparatively fewer studies. Of the 1981 unique stations that have been sampled for <sup>234</sup>Th (Ceballos-Romero et al., 2022), only 1.4 % are in nearshore coastal settings. Importantly, only few of the 223 studies make coupled, quasi-simultaneous measurements of the <sup>234</sup>Th deficit in the water column and the expected excess of <sup>234</sup>Th in surficial sediments (Cochran et al., 1995; Charette et al., 2001; Black et al., 2023; Lepore et al., 2007; Turnewitsch and Springer, 2001). By using this approach, detailed insights into carbon export can be made that aid in carbon budget evaluations. This study applies the coupled water column-sediment approach to evaluate both <sup>234</sup>Th and POC sinking fluxes and their respective sediment accumulation fluxes in the Bedford Basin, a coastal fjord in the Northwest Atlantic Ocean.

#### 2. Methods








#### 2.1. Study site

The Bedford Basin (BB; 44.6955° N, 63.6330° W; Fig. 1) is a 17 km<sup>2</sup>, semi-enclosed fjord in Nova Scotia, Canada, and forms the northwestern part of the Halifax Harbour. The basin has a volume of 5.6 x 108 m<sup>3</sup> (Burt et al., 2013) and a maximum depth of 70 m. It is connected to the inner Halifax Harbour via a 20 m deep sill and to the open Atlantic Ocean by a 10 km long channel that is less than 400 m wide at its narrowest point (Li and Dickie, 2001). Runoff from the Sackville River (1.5 × 10<sup>8</sup> m<sup>3</sup> yr<sup>-1</sup>, Kepkay et al., 1997) and surrounding areas is discharged into the basin, creating a two-layer estuarine circulation – a fresher top layer and a denser bottom layer from the shelf water intruding from the entrance of the Halifax Harbour. The hydrography of the basin is characterized by annual cycles of winter mixing and summer stratification. Stratification is established in the spring (April–May) and persists into the fall and winter (November–February). In most years, the water column fully mixes during the late winter (mid-February to mid-March). Seasonal hypoxia occurs in the deeper basin during the fall, but re-oxygenation takes place during late winter mixing. In spring, a phytoplankton bloom typically occurs as surface stratification increases. Chlorophyll-a (chl-a) concentrations in surface waters (1–10 m) typically range from ~3–10 mg m<sup>-3</sup>. Spring blooms typically range from ~7–28 mg m<sup>-3</sup> (Fisheries and Oceans Canada, 2025), while extreme short-lived events can exceed 100 mg m<sup>-3</sup> (Li and Dickie, 2001). Occasional intrusions of dense Scotian Shelf water can replenish either the bottom or mid-depth waters (Shi and Wallace, 2018; Haas et al., 2021). The Bedford Basin Monitoring Program (BBMP), established by the Bedford Institute of Oceanography (BIO) has conducted weekly measurements of water properties since 1992. Despite a large array of oceanographic variables that are frequently sampled by BBMP and other emerging research initiatives (Fennel et al., 2023; Burt et al., 2024), limited information exists on POC export and burial in this important urban coastal region.

*Figure 1*: A bathymetric map of the Bedford Basin (GEBCO Gridded Bathymetry Data). Compass Buoy Station (70 m), the main sampling location, is represented by the yellow star. The inset map (upper right corner) shows the relative position of the Bedford Basin in Halifax, Nova Scotia, Atlantic Canada.

# 2.2. Hydrographic properties









Weekly monitoring at the Compass Buoy Station in the centre of the Bedford Basin provides temperature, salinity, oxygen, and chl-a concentrations (BBMP, Seabird CTD SBE-25). To capture the mean conditions near the basin floor, we derived mean values for temperature, salinity, and oxygen between 60 m water depth and the bottom (or the deepest available measurement). We then calculated the anomalies between consecutive profiles, resulting in a time series of anomalies that reflects changes from week to week in each variable in the deep basin (Fig. 2d).

## 2.3. Total <sup>234</sup>Th and <sup>238</sup>U activities in seawater

Total (dissolved + particulate) <sup>234</sup>Th (<sup>234</sup>Th<sub>tot</sub>) was measured six times at the Compass Station between 2021–2024. Water column samples were collected using 10 L Niskin bottles at 9 discrete depths. From each bottle, a 2 L subsample of unfiltered seawater was taken, spiked with a <sup>230</sup>Th yield tracer, and processed according to the MnO<sub>2</sub> co-precipitation technique (Pike et al., 2005). Following co-precipitation, samples were filtered onto quartz fiber filters (QMA, 25 mm diameter, 1 µm nominal pore) following Cai et al. (2006). After filtration, filters were dried (55° C), mounted on sample holders, covered with Mylar<sup>TM</sup> film (1 layer) and aluminium foil (2 layers), and their beta emissions were measured on low-level beta GM multi-counters (Risø National Laboratory, Denmark). Samples were counted until the error was typically < 3 %. After > 120 days from collection (i.e., after 5-6 half-lives of <sup>234</sup>Th had passed), samples were recounted to determine the non <sup>234</sup>Th beta activity stemming from other radionuclides included in the precipitate, which was subtracted from the first count. After final counting was completed, the chemical recovery of <sup>230</sup>Th was determined by inductively coupled plasma mass spectrometry (ICP-MS) as outlined in Clevenger et al. (2021), yielding a mean recovery of 85 ± 12 % (n = 63). The net counting rate was corrected for <sup>234</sup>Th decay and ingrowth, counting efficiency, and chemical recovery. The uncertainties in the <sup>234</sup>Th<sub>tot</sub> activities are determined from counting statistics and the propagation of errors in relation to sample processing, including mass and volume measurements, and ICP-MS recovery analysis. The parent <sup>238</sup>U activity is derived from salinity following the relationship given by Owens et al. (2011), (Eq. 1).

$$^{238}U \ (\pm 0.047 \ \mathrm{dpm} \ L^{-1}) = (0.0786 \ \pm 0.045) \ \times \ Salinity = (0.315 \ \pm 0.158)$$
 (1)

Additional work conducted on 15 January 2025 measured salinity (PSU) and dissolved  $^{238}$ U activities at multiple locations and depths in the Bedford Basin across a salinity range of 0.1–31.2 (n = 26) and yielded a strong correlation with salinity (R<sup>2</sup> = 0.99). At the Compass Station specifically, measured dissolved  $^{238}$ U activities (at 1, 6 and 50 m) were within 1% of the activities predicted by the U-Salinity relationship, confirming the applicability of the Owens et al. (2011) relationship at this site.

Seawater <sup>234</sup>Th activities are used to calculate cumulative water column <sup>234</sup>Th fluxes (dpm m<sup>-2</sup> d<sup>-1</sup>) as follows:

$$Flux_{Th@z} = \int_{0}^{z} (\lambda_{Th}(^{238}U - ^{234}Th) + w(\frac{\partial^{234}Th}{\partial z}) - u(\frac{\partial^{234}Th}{\partial z}) - v(\frac{\partial^{234}Th}{\partial z})) dz$$
 (2)

Here,  $^{234}$ Th is  $^{234}$ Th<sub>tot</sub> activity (dpm  $L^{-1}$ ),  $^{238}$ U is the salinity derived  $^{238}$ U activity (dpm  $L^{-1}$ ; Owens et al., 2011),  $\lambda_{Th}$  is the decay constant of  $^{234}$ Th (0.0288 d<sup>-1</sup>), w is the upwelling velocity, u is the zonal velocity, v is the meridional velocity, and the  $\partial$  terms are the vertical ( $\partial z$ ), west to east ( $\partial x$ ), and south to north ( $\partial y$ ) gradients. Applying the 1D steady state model, Eq. 2 can be simplified to:

$$Flux_{Th@z} = \int_0^z (\lambda_{Th}(^{238}U - ^{234}Th) dz$$
 (3)

Given the temporal resolution of this study, and the fact that <sup>234</sup>Th profiles were not measured during consecutive weeks—months but rather on a quasi-seasonal scale, the use of a non-steady-state model for <sup>234</sup>Th flux is not feasible, and we apply the steady state condition. The validity of the 1D steady-state assumption is discussed further in Section 4.3. The uncertainties of the flux are the propagated measurement error.

## 2.4. Particulate <sup>234</sup>Th and POC in seawater

Size fractionated particulate <sup>234</sup>Th samples (<sup>234</sup>Th<sub>p</sub>) were collected by in-situ pumping of ~ 30 L at 3 depths (~20, 40, 60 m) using large volume pumps (McLane Laboratories<sup>TM</sup>). Filter heads were equipped with a 51 μm pore size Nitex filter above a 1 μm pore size QMA filter (pre-combusted). The large particle material (> 51 μm) captured on the Nitex screens was carefully rinsed onto QMA filters (1 μm pore size, 25 mm diameter, pre-combusted) using 0.2 μm filtered and ultraviolet (UV) treated seawater. The small particulate material (1–51 μm) captured on the QMA filter was subsampled by cutting out a 25 mm diameter punch. After drying (55 °C) all particulate samples were mounted onto sample holders and analyzed on beta counters (see above). Following the second counting, particulate samples were prepared for POC measurement by fumigation under HCl vapor for 6 h inside a glass desiccator to remove the carbonate phase. Samples were dried at 50 °C and packed into tin cups and then measured on an Elemental Analyzer (Elementar MicroCube & Isoprime 100). For each pump cast, an additional filter head was attached to the deepest pump to obtain a seawater process blank ("dipped blank"; Lam et al., 2015). The dipped blanks were exposed to seawater for the same amount of time and processed as regular samples. Dipped blank filters were used as process blanks for blank subtraction, detection limit determination, and uncertainty calculations. The POC flux can be calculated with either (a) the POC:<sup>234</sup>Th ratio on small particles, or (b) the POC:<sup>234</sup>Th ratio on large particles using Eq. 4. The fluxes reported using either (a) or (b) represent two alternative approaches to estimating flux, and should not be considered additive, but rather complementary perspectives on export.

$$POC Flux = Flux_{Th @ z} \times \left(\frac{POC}{^{234}Th}\right)_{(a),(b)}$$
 (4)

To investigate the role of resuspension in this study and in the Bedford Basin as a whole, the residual beta activity of particulate  $^{234}$ Th (Ra<sub>P234</sub>) was analyzed. Ra<sub>P234</sub> is a novel resuspension proxy (Lin et al., 2016) and has previously been applied to investigate particle dynamics in coastal sites. Briefly, the Ra<sub>P234</sub> signal comes from "supported" radionuclides bound to particle components like biogenic or lithogenic materials. Terrigenous particles retain radionuclides such as  $^{234}$ Th (from  $^{238}$ U decay) and  $^{212}$ Bi (from  $^{228}$ Th decay), contributing to  $\beta$  counting due to their long half-lives. In contrast, "unsupported"  $^{234}$ Th is adsorbed in the water column and is the focus of this (and other studies) using the  $^{234}$ Th/ $^{238}$ U method to quantify carbon fluxes. The supported Ra<sub>P234</sub> signal indicates sediment resuspension, potentially biasing particle flux measurements. Ra<sub>P234</sub> was obtained from the second counting rates of  $^{234}$ Th<sub>p</sub> after 5-6 half-lives (> 120 d) passed, and calculated as:

$$RA_{P234} = \left(\frac{n_{p2} - n_0}{\epsilon V}\right) \tag{5}$$

Where  $n_{p2}$  is the second  $\beta$  counting rate of  $^{234}$ Th<sub>p</sub> (mean  $0.3 \pm 0.07$  counts per minute (cpm)),  $n_0$  is the blank mean ( $0.24 \pm 0.02$  cpm),  $\epsilon$  is the detector efficiency (0.38), and V is volume of seawater pumped (L). The detection limit is defined here as two times the standard deviation of the blank and corresponds to 128.2 dpm m<sup>-3</sup>.

#### 2.5. Excess <sup>234</sup>Th inventories in seafloor sediments









For the measurement of <sup>234</sup>Th in the sediment, a KC Denmark<sup>TM</sup> Multicorer was deployed at the Compass Station following water column sampling. Sediments were processed either the same day of retrieval or stored at (4° C) for 24–48 h prior to processing. Cores were sliced in 0.5–1 cm thick sections and then dried at 55° C for approximately 24 h. After this period, they were weighed and returned to the oven and reweighed at regular intervals until a constant weight was achieved. To correct for residual salt content, salt contribution from porewater was estimated from the salinity of the overlying water and subtracted from the measured dry weight. Once dry, sediments were manually homogenized and weighed into a petri-dish. Analyses of sedimentary <sup>234</sup>Th and <sup>238</sup>U activities were carried out by non-destructive gamma counting on High-Purity Germanium (HPGe) detectors (Dartmouth College, USA) with ~ 15 g of sediment. The raw <sup>234</sup>Th activities (at 63.3 keV) in sediments were decay corrected to time of collection. Unsupported or 'excess' <sup>234</sup>Th (<sup>234</sup>Th<sub>xs,0</sub>) is <sup>234</sup>Th that is not in secular equilibrium with the <sup>238</sup>U parent present in the uppermost sediments, presumably because it was supplied by settling particles. Supported <sup>234</sup>Th, in equilibrium with <sup>238</sup>U present in the sediment, is identified at the core depth below which no further decrease in <sup>234</sup>Th activities is observed ("Equilibrium Depth"; EQ). The supported activity is subtracted from the total activity to derive <sup>234</sup>Th<sub>xs,0</sub> (Eq. 6).

$$^{234}Th_{xs,0} = (Decay\ Corrected \quad ^{234}Th) - (Supported \quad ^{234}Th)\ (dpm\ g^{-1}) \tag{6}$$

Sediment inventories of  $^{234}$ Th<sub>xs,0</sub> were multiplied by dry bulk density, and the decay constant of  $^{234}$ Th (0.0288 d<sup>-1</sup>) and integrated from 0 to the EQ to derive the sediment accumulation flux of  $^{234}$ Th<sub>xs,0</sub> on the sea floor (Eq. 7).

Finally, to derive the sediment accumulation flux of POC, following Eq. 4, the  $^{234}$ Th sediment accumulation flux (Eq. 7) are multiplied by the water column (a) small and (b) large particle POC: $^{234}$ Th ratios measured at 65 m, the depth which represents particles that are about to settle on the seafloor (70 m). The POC: $^{234}$ Th ratios directly measured in the sediment are not used as  $^{234}$ Th decays away faster than organic carbon is respired in sediments. Total organic carbon (TOC) content in sediments was measured on an Elemental Analyzer (Elementar MicroCube) at Dalhousie University. Previous work on Bedford Basin sediments compared acid-fumigated and non-acid fumigated samples and found that the mean carbon content did not differ between the acid fumigated (5.5 %  $\pm$  0.5 %, n=29) and non-acid fumigated (5.7 %  $\pm$  0.8 %, n=39) samples (Black et al., 2023). This published finding is consistent with additional unpublished measurements that were made over the years at Dalhousie University. Collectively, the data imply that most of the sedimentary carbon is organic in nature, and thus acid fumigation was not done here.

Sediment Accumulation Flux 
$$^{234}Th = \int_0^{EQ} \lambda Th \left(^{234}Th - ^{238}U\right) dz = \int_0^{EQ} \lambda \left(^{234}Th_{xs,0}\right) dz$$
 (7)

#### 3. Results

#### 3.1. Hydrography of Bedford Basin in 2021 - 2024

Hydrographic conditions in the Bedford Basin (2021–2024) generally align with long term trends (Li and Harrison, 2008; Haas et al., 2021). Winter and spring surface cooling reduce summer stratification and drive vertical convection (Fig. 2a). However, 2021 saw higher sea surface temperatures (SSTs), leading to stronger stratification (Rakshit et al., 2023), and a lack of shelf water intrusions (Fig. 2e). That year, bottom waters experienced a six-week oxygen minimum, in contrast with records from 2014–2020 when mixing and intrusions increased dissolved oxygen in the Bedford Basin (Haas et al., 2021). The absence of winter mixing in 2021 may be linked to warmer air temperatures from January–March 2021 (- 0.2 °C), compared to the long-term mean (Environment and Climate Change Canada, 2018) at this site (- 1.5 °C). A spring phytoplankton bloom followed the cessation of winter mixing (Fig. 2d), with chlorophyll maxima at 5–10 m depth. The 2021 bloom, as denoted by chl-a signals, was relatively weak, (11.1 mg m<sup>-3</sup>) compared to stronger blooms observed in 2022 and 2023. The largest spring bloom sampled in this study occurred in 2023 (34.9 mg m<sup>-3</sup>). Deep chl-a signals are observed down to 70 m depth in some sampling events (May 2023). A secondary fall bloom occasionally occurred, including one in late summer 2021 (15.4 mg m<sup>-3</sup> chl-a). Two intrusion events were observed throughout 2021–2024 (Nov 2022, January 2024; Fig. 2e).

Figure 2: Hydrographic conditions in Bedford Basin (Compass Buoy Station) from 2021–2024. Time series of (a) Temperature (°C), (b) Salinity (PSU), (c) Oxygen concentration (μM), (d) Chl-a concentration (mg m<sup>-3</sup>), (e) anomalies of temperature, oxygen concentration and salinity (T,S,O), which trace the intrusion of shelf waters into the basin. A zero anomaly

represents no variation between T,S,O between weeks. The grey shaded area represents times likely of an intrusion event. Dashed lines in all panels represent the sampling events in this study.

## 3.2. Total <sup>234</sup>Th in the water column





Throughout the water column,  $^{234}$ Th<sub>tot</sub> activities displayed large deficits ( $^{234}$ Th<sub>tot</sub> = 0.1–1.1 dpm L<sup>-1</sup>) compared to  $^{238}$ U activities (mean =  $^{2.09}$  ± 0.04 dpm L<sup>-1</sup>) at all sampling events and all depths (Fig. 3). This indicates that the particle flux in Bedford Basin was large enough in the entire water column to scavenge  $^{234}$ Th prior to its decay. Similar observations have been made in the few other studies from coastal regions (e.g., Luo et al., 2014; Wei and Murray, 1992; Black et al., 2023; Lepore et al., 2007). This contrasts with open ocean sites where  $^{234}$ Th deficits are typically restricted to the euphotic zone (Owens et al., 2015). Generally, in each season, consistency is seen between years, with some variation that could be due to differences in bloom timing, productivity, etc. The largest deficit, with  $^{234}$ Th activities as low as 0.1 dpm L<sup>-1</sup>, occurred during the large bloom event in May 2023 (Fig. 2d), suggesting enhanced scavenging during this time. Winter profiles (Fig. 3c) are more uniform with depth compared to spring and summer, and despite low productivity during these times (Fig. 2d), still show a pronounced deficit throughout the entire water column.

*Figure 3:* Water column profiles of total <sup>234</sup>Th (dpm L<sup>-1</sup>) activity in each season sampled. (a) Spring (April, May), (b) Summer (August), (c) Winter (January) compared to <sup>238</sup>U activity (dpm L<sup>-1</sup>) derived from salinity (Owens et al., 2011).

## 3.3. Size fractionated particulate <sup>234</sup>Th and POC activities in water column

Along with <sup>234</sup>Th<sub>tot</sub>, size fractionated particulate <sup>234</sup>Th (<sup>234</sup>Th<sub>p</sub>) were obtained at approximately 20, 40, 60 m depths by in-situ pumps. At all depths, the small (1–51 μm) particle fraction dominated the <sup>234</sup>Th<sub>p</sub> activities (93%) and POC concentration (77.4%), suggesting a large amount of POC is suspended on particles in the water column (Fig. 4a). POC on the small particle fraction had a mean of  $4.5 \pm 3.1 \,\mu\text{M}$ , while POC on the large particle fraction had a mean of  $1.0 \pm 1.2 \,\mu\text{M}$ . Mean  $^{234}\text{Th}_{\text{D}}$  activities on the small size fraction  $(0.8 \pm 0.3 \text{ dpm L}^{-1})$  were higher than those on large particles. Mean <sup>234</sup>Th<sub>n</sub> activities on large size 280 fraction particles were  $0.06 \pm 0.04$  dpm  $L^{-1}$  (Fig. 4b) and were highest during bloom events. This value is significantly larger than first measurements of  $^{234}$ Th<sub>p</sub> in the Bedford Basin (0.008  $\pm$  0.006 dpm L<sup>-1</sup>, Black et al., 2023), likely due to a difference in methods (see Section 4.4). Although dissolved <sup>234</sup>Th (<sup>234</sup>Th<sub>diss</sub>) was not measured in this study, subtracting <sup>234</sup>Th<sub>p</sub> from the <sup>234</sup>Th<sub>tot</sub> at the same approximate depth provides an estimation. The activity of dissolved <sup>234</sup>Th in Bedford Basin is consistently low and nearly devoid (0-0.6 dpm L<sup>-1</sup>), a finding that is similar in other coastal fjords (e.g. Luo et al., 2014). Hence, the majority of <sup>234</sup>Th<sub>tot</sub> is in the <sup>234</sup>Th<sub>p</sub> pool, consistent with other studies in the Bedford Basin (Black et al., 2023; Niven et al., 1995). The 285 residual activity of particulate <sup>234</sup>Th (Rap<sub>234</sub>) reveal that residual beta counts were higher than the detection limit in 38 % of the small particulate samples, but no counts exceeding the detection limit were found on the large particles (Fig. 9).


*Figure 4:* Size-fractionated (1–51  $\mu$ m, >51  $\mu$ m) pools of (a) particulate organic carbon ( $\mu$ M) and (b) total particulate <sup>234</sup>Th (dpm L<sup>-1</sup>) at approximately 20, 40, and 60 m for each sampling event. Bars are stacked and totals equal the sum of the small and large fractions (legend shows % contribution by size class). Numbers above bars indicate approximate sampling depth. Dashed lines separate events. The small fraction generally dominates both POC and <sup>234</sup>Th

# 295


# 3.4. POC:<sup>234</sup>Th ratios on small and large particles

In general, POC: $^{234}$ Th on both the large and small particle fractions are  $

Figure 5: Water column ratios of POC to <sup>234</sup>Th (POC:<sup>234</sup>Th; μmol C dpm<sup>-1</sup>) in each sampling event throughout 2021–2024. Error bars indicate propagated uncertainties associated with measurements. (a) POC:<sup>234</sup>Th on the small particle (1–51 μm) fraction (b) POC:<sup>234</sup>Th on the large particle (> 51 μm) fraction. Other than a few large POC:<sup>234</sup>Th ratios that correspond with greater chl-a in the water column (Fig. 2d), the POC:<sup>234</sup>Th ratios are similar in magnitude on both size fractions (< 20 μmol C dpm<sup>-1</sup>; grey reference line). The dashed grey line at 70 m represents the bottom of the Bedford Basin.

# 3.5. Water column fluxes of <sup>234</sup>Th and POC

## 3.5.1. <sup>234</sup>Th fluxes





Cumulative  $^{234}$ Th fluxes increase nearly linearly with water depth ( $R^2 = 0.95$ ) and do not show much seasonal variation (Fig. 6a). Interestingly, similar observations were made in other coastal sites such as Dabob Bay, Washington (Wei and Murray, 1992), Saanich Inlet, British Columbia (Luo et al., 2014), Beaufort Sea Shelf (Baskaran et al., 2003), and Mecklenburg Bay, Baltic Sea (Forster et al., 2009). Combining the data from the linear regression of all dates reveals a water column rate increase with depth of 37 dpm m<sup>-2</sup> d<sup>-1</sup>, larger than previously measured in spring and summer 2019 (31 dpm m<sup>-2</sup> d<sup>-1</sup>; Black et al., 2023). The mean ( $\pm$  s.d.) water column  $^{234}$ Th fluxes at  $\sim$  65 m and extrapolated to 70 m were  $2638 \pm 238$  and  $2842 \pm 224$  dpm m<sup>-2</sup> d<sup>-1</sup> respectively.

#### 3.5.2. POC fluxes derived with the small and large particle size fraction

POC export fluxes (Fig. 6b, Fig. 6c) were calculated following Eq. 4 using both the small (1–51  $\mu$ m) and large (> 51  $\mu$ m) POC:<sup>234</sup>Th ratios (Fig. 5a and 5b, respectively). Fluxes based on the small size fraction showed no consistent pattern with depth, while fluxes on the large size fraction varied more strongly. Small particle fluxes at 65 m had a mean ( $\pm$  s.d.) of 12.5  $\pm$  7.1 (Fig. 7a) and ranged from 0.8 to 19.6 mmol C m<sup>-2</sup> d<sup>-1</sup> (Fig. 6b). In general, large particle fluxes followed a high-low pattern with depth (Martin et al., 1987), and fluxes at 65 m had a mean ( $\pm$  s.d.) of 20  $\pm$  14 mmol C m<sup>-2</sup> d<sup>-1</sup> (Fig. 7c) and ranged from 3.6  $\pm$  1.1 to 44.5  $\pm$  4.1 mmol C m<sup>-2</sup> d<sup>-1</sup>. Highest POC flux values were seen in May when chl-a was at its highest (Fig. 6c). Modest POC fluxes were still prevalent in boreal winter estimates, averaging 15.7  $\pm$  7.2 mmol C m<sup>-2</sup> d<sup>-1</sup>. One of these winter sampling events occurred around the time of an intrusion event in the Bedford Basin, and it is possible that this event brought additional POC from nearby shelf waters (see Section 4.2). Disregarding this sampling event, POC fluxes in January 2022 were 10.6  $\pm$  1.5 mmol C m<sup>-2</sup> d<sup>-1</sup>.

Figure 6: Bedford Basin 1D steady state fluxes in 2021–2024. (a) Profiles of cumulative  $^{234}$ Th flux in the water column compared to decay corrected excess  $^{234}$ Th ( $^{234}$ Th<sub>xs,0</sub>) in the topmost sediment. Sediment inventories of  $^{234}$ Th<sub>xs,0</sub> were integrated from ~ 0–EQ depth to derive accumulation of  $^{234}$ Th<sub>xs,0</sub> on the sea floor (Eq. 7). The resulting sediment fluxes (i.e. sediment accumulation flux) are shown below the approximate depth of the sediment-water interface (dashed line). POC fluxes on (b) the small (1–51 μm) particle fraction, and (c) large (> 51 μm) particle fraction. The sediment accumulation flux was also multiplied by the water column POC: $^{234}$ Th ratios (at 60–65 m) to derive POC sediment accumulation flux. Note the change in scale between the plots and the grey reference line at 40 mmol C m<sup>-2</sup> d<sup>-1</sup>. No sediment core was taken in August 2023.

Figure 7: Depositional POC fluxes to the seabed in the Bedford Basin. Measured <sup>234</sup>Th-derived mean (± s.d.) particulate organic carbon fluxes deposited to the near bottom (~ 65 m) of the Bedford Basin on the (a) small particle (1–51 μm) and (b) large particle (> 51 μm) size fractions. The dashed black line represents the mean POC flux for each size fraction respectively.

**Table 1.** Water column <sup>234</sup>Th cumulative fluxes measured at the bottom of the Bedford Basin (60–65 m). Water column <sup>234</sup>Th fluxes are multiplied by the POC:<sup>234</sup>Th ratios measured on small and large particles to derive POC fluxes with a steady state model. Note that the sampling events with an asterisk (\*) represent data from a pilot study in 2019 (Black et al., 2023).

| Sampling     | Sampling   | Season | Water                  | Small                  | Small                  | Large                  | Large                  |
|--------------|------------|--------|------------------------|------------------------|------------------------|------------------------|------------------------|
| Event        | Date       |        | Column                 | particle               | particle               | particle               | particle               |
|              |            |        | <sup>234</sup> Th flux | POC: <sup>234</sup> Th | POC flux               | POC: <sup>234</sup> Th | POC flux               |
|              | M-D-Y      |        | $dpm m^{-2} d^{-1}$    | μmol C                 | mmol C m <sup>-2</sup> | μmol C                 | mmol C m <sup>-2</sup> |
|              |            |        |                        | $dpm^{-1}$             | $d^{-1}$               | $dpm^{-1}$             | $d^{-1}$               |
| April 2019*  | 4-11-2019  | Spring | $2140\pm160$           | 11 ± 4                 | $25 \pm 10$            | $340\pm10$             | $780 \pm 190$          |
| May 2019*    | 5-09-2019  | Spring | $1890\pm150$           | $11 \pm 6$             | $20\pm10$              | $920 \pm 4$            | $2070 \pm 500$         |
| April 2021   | 04-27-2021 | Spring | $2205 \pm 64$          | $7.1 \pm 1$            | $15.1\pm1.2$           | $9.1 \pm 1$            | $20.1\pm2.3$           |
| May 2023     | 05-05-2023 | Spring | $2767 \pm 62$          | $0.3\pm0.07$           | $0.8 \pm 0.2$          | $67 \pm 8.3$           | $44.5 \pm 4.1$         |
| August 2019* | 08-02-2019 | Summer | $2020 \pm 150$         | $60 \pm 20$            | $150 \pm 40$           | $300\pm80$             | $710\pm170$            |
| August 2021  | 08-23-2021 | Summer | $2537 \pm 61$          | $7.7 \pm 1$            | $19.6 \pm 3.0$         | $8.1 \pm 1$            | $20.5 \pm 2.1$         |
| August 2023  | 08-02-2023 | Summer | $2726\pm50$            | $2.9 \pm 0.4$          | $8\pm0.6$              | $1.3 \pm 0.6$          | $3.6\pm1.7$            |
| January 2022 | 01-05-2022 | Winter | $2603 \pm 46$          | $7.3 \pm 1.2$          | $19 \pm 3.1$           | $4.1 \pm 0.6$          | $10.6 \pm 1.5$         |
| January 2024 | 01-31-2024 | Winter | $2478 \pm 54$          | $4.9 \pm 0.9$          | $12.2\pm2.1$           | $8.3 \pm 0.7$          | $20.8\pm1.5$           |

# 3.5.3. Sediment accumulation fluxes of <sup>234</sup>Th and POC

350

Excess <sup>234</sup>Th was concentrated in the surface sediments, with the highest activities observed in the top 1.5 cm (Fig. 8a). Excess <sup>234</sup>Th decreased with depth, and equilibrium with supported <sup>234</sup>Th was consistently reached by 4.5–6 cm in every core. Excess

<sup>234</sup>Th above the EQ was variable in each sampling event, averaging  $15.3 \pm 16$  dpm g<sup>-1</sup> and ranging from 4.8–46 dpm g<sup>-1</sup> (Fig. 8a). The mean supported <sup>234</sup>Th (from 4.5–6 cm) was  $2 \pm 0.5$  dpm g<sup>-1</sup>. The mean sedimentary POC concentration was  $4.9 \pm 0.4$  mmol C g<sup>-1</sup>, with an average (wt %) of  $5.7 \pm 0.3$  % TOC. POC concentrations were consistent both downcore (0–6 cm) and between cores (n=5), in line with previous studies in the Bedford Basin (Black et al., 2023; Buckley et al., 1995). The POC:<sup>234</sup>Th ratios in sediments were an order of magnitude higher than those in the overlying water column, ranging from 103–3257 μmol dpm<sup>-1</sup> (Fig. 8b). This difference likely reflects the more rapid decay of <sup>234</sup>Th relative to the remineralization of OC on the seabed. These ratios increased with core depth, corresponding with decreasing <sup>234</sup>Th<sub>xs,0</sub> activity (Fig. 8b). The sediment accumulation fluxes of <sup>234</sup>Th had a mean (± s.d.) of  $2450 \pm 549$  dpm m<sup>-2</sup> d<sup>-1</sup> and ranged from  $1735 \pm 237$  to  $3211 \pm 249$  dpm m<sup>-2</sup> d<sup>-1</sup> (Table 2). The <sup>234</sup>Th sediment accumulation fluxes were multiplied by the POC:<sup>234</sup>Th ratio of water column particles collected at the bottom of the basin (60–65 m) to derive POC sediment accumulation fluxes. The mean (± s.d.) POC sediment accumulation flux derived with the small particle fraction was  $13 \pm 8$  mmol C m<sup>-2</sup> d<sup>-1</sup>, lower than the mean POC flux derived with the large particle fraction of  $22 \pm 9$  mmol C m<sup>-2</sup> d<sup>-1</sup> (Fig 6b, Fig 6c).

*Figure 8:* Sediment downcore profile of (a) decay corrected excess  $^{234}$ Th ( $^{234}$ Th<sub>xs,0</sub>), derived from subtracting the supported  $^{234}$ Th from decay corrected  $^{234}$ Th in sediments. (b) The ratio of POC: $^{234}$ Th<sub>xs,0</sub> in sediments.

Table 2. Bedford Basin sediment accumulation fluxes of  $^{234}$ Th (dpm m $^{-2}$  d $^{-1}$ ) in each sampling event in 2021–2024, calculated by multiplying the integrated activities of  $^{234}$ Th $_{xs,0}$  (0–EQ depth) by dry bulk density and the decay constant for  $^{234}$ Th. Note that there was no sediment sampling in August 2023.

| Sampling Event | Sampling Date (M-D-Y) | Season | <sup>234</sup> Th sediment accumulation flux (dpm m <sup>-2</sup> d <sup>-1</sup> ) |
|----------------|-----------------------|--------|-------------------------------------------------------------------------------------|
| April 2021     | 04-27-2021            | Spring | $2687 \pm 184$                                                                      |
| May 2023       | 05-05-2023            | Spring | $1907 \pm 347$                                                                      |
| August 2021    | 08-23-2021            | Summer | $2713 \pm 239$                                                                      |
| January 2022   | 01-05-2022            | Winter | $1735 \pm 237$                                                                      |
| January 2024   | 01-31-2024            | Winter | $3211 \pm 249$                                                                      |

#### 4. Discussion







# 4.1. A general match between water column <sup>234</sup>Th flux and surficial sediment <sup>234</sup>Th accumulation flux

This study demonstrates the utility of coupled water column-sediment <sup>234</sup>Th measurements in assessing coastal carbon fluxes using a 1D steady-state model. To the best of our knowledge, few studies have applied a coupled water column-sediment approach to explore the <sup>234</sup>Th/<sup>238</sup>U method in support of coastal carbon budget quantifications (Cochran et al., 1995; Charette et al., 2001; Lepore et al., 2007; Turnewitsch and Springer, 2001). The present study, along with the first coupled measurements in Bedford Basin made in 2019 (Black et al., 2023), demonstrates a close match between the flux at the bottom of the water column and the uppermost sediments beneath it. Overall, the mean ( $\pm$  s.d.) 2021–2024 sediment accumulation of <sup>234</sup>Th of 2261  $\pm$  512 matches within error to the mean water column flux (interpolated to 70 m) of 2851 ± 249 (Fig. 6a). A paired t-test revealed no statistically significant difference between the water column and sediment fluxes, with a mean difference ( $\pm$  s.d.) of 349  $\pm$  747 dpm m<sup>-2</sup> d<sup>-1</sup> (t = 1.04; df= 4, p=0.30). When data from 2019 (Black et al., 2023) are included, the mean difference ( $\pm$  s.d.) is even smaller (184.1  $\pm$  836) and not statistically significant (t = 0.6; df=7, p=0.5), suggesting that the observed difference may be due to random variability. The mean ratio between water column and sediment <sup>234</sup>Th fluxes is 1.1 indicating close agreement between the two, with the water column flux being slightly higher than the sediment flux. This finding implies that on the timescales of weeks to ~ 100 days, no major particle loss is occurring, and particles are generally not being exported out of the basin. A similar finding was first made by Cochran et al. (1995) who found a water column-sediment inventory match within a factor of 2 in both 1992 and 1993 on the Northeast Water Polynya off Greenland. Likewise, in their comparison of model predicted <sup>234</sup>Th water column inventory vs sediment inventories in the upper 0–5 cm of the Gulf of Maine, Charette et al. (2001) find that inventories were within a factor of 2 of the predicted values, with a ratio of 1.1. These findings suggest that Bedford Basin is a promising location for more complex observational studies of particle dynamics, and opens the opportunity to study other short-lived radioisotopes, such as Bismuth-210 ( $^{210}$ Bi;  $t_{1/2} = 5$  d) to study particle dynamics on a finer temporal resolution (hours to days; see Yang et al., 2022).

## 4.2. A persistent year-round deficit of <sup>234</sup>Th in the water column

A notable feature of this study is the persistent deficit of <sup>234</sup>Th relative to <sup>238</sup>U in all seasons sampled. Similar findings have been reported in other shallow basins (Luo et al., 2014; Charette et al., 2001; Evangeliou et al., 2011; Amiel and Cochran, 2008). This persistent deficit suggests that i) particle scavenging of <sup>234</sup>Th is occurring throughout the entirety of the water column (0–70 m)

and ii) the scavenging rate is high enough to continually maintain a lower <sup>234</sup>Th activity relative to <sup>238</sup>U. Particles in the Bedford Basin may originate from both biological processes in the upper 30 m (Fig. 3), and from resuspension.








Resuspension from the seafloor could be a source of particles to the water column, though multiple lines of evidence indicate it is likely minimal at the Compass Station. First, sedimentary POC:<sup>234</sup>Th ratio is an order of magnitude higher than those in the water column (Fig 8b), likely due to the temporal decay of <sup>234</sup>Th exceeding the remineralization of POC. Bottom resuspension would be expected to introduce particles with these elevated POC:<sup>234</sup>Th which is not reflected in deep water column POC:<sup>234</sup>Th ratios from this study. Second, recent measurements of physical properties in the Bedford Basin report very weak near-bottom currents (typically < 1–2 cm s<sup>-1</sup>), even during tidal cycles at the Compass Station. Other coastal studies in shallow systems have indicated that near-bed velocities of 20–30 cm s<sup>-1</sup> are generally required to erode bottom sediments (Ziervogel et al., 2021). These findings support negligible resuspension from the seafloor at the Compass Station. However, given the concave bathymetric profile of the Bedford Basin (Fig. 1), resuspended sediments could be "funneled" into the Compass Station from the sides (Trimble and Baskaran, 2005). In this study, we observe a strong signal of resuspension at mid depths in April 2021, as well as a weaker signal of resuspension at 60 m in the summer and winter sampling events (Fig. 9), though only on the small particle fraction. This Rap234 signal provides indication that some lateral resuspension of fine particles is likely occurring in the Bedford Basin and is reason to report POC fluxes using the large particle fraction. Furthermore, deep and extensive CTD-derived chl-a fluorescence signals (Fig. 2d) suggest that biological processes could also be responsible for particle creation and continued scavenging in deeper waters. These features are not unique (Black et al., 2023), and fluorescent organic material has been visually observed at depths > 3000 m in the Arctic as a result of a rapid bloom-collapse cycle (Boetius et al., 2013). While additional data (e.g. Underwater Vision Profilers (UVP), net primary production (NPP), backscatter, etc.) are needed to identify the source for the evident fluorescence signals at depths much deeper than expected given the high particle load (i.e. elevated chl-a and POC: <sup>234</sup>Th) in the water column, these features could be the key to understanding the persistent Th deficits in the Bedford Basin.

The persistent deficit is also strong in boreal winter periods, when production is presumed to be at its lowest. As argued by Hargrave and Taguchi (1978), inflows of dense surface water from intrusions could deliver additional organic material that could drive the moderate POC fluxes we see at this time. Therefore, the POC flux in January 2024 could be attributed to the intrusion event occurring around the same time. However, our sampling in January 2022 still yields a modest flux value of  $10.6 \pm 1.5$ mmol C m<sup>-2</sup> d<sup>-1</sup> at 65 m. This is comparable to January 2024 (Fig. 7) but took place after a period of water column stability (Fig. 2e). A notable feature of this 2022 winter event is that POC fluxes are higher when using the small fraction to compute flux, yielding  $17.7 \pm 1.3$  mmol C m<sup>-2</sup> d<sup>-1</sup> (Fig. 7a). Small phytoplankton (< 20 µm) can play an important role in carbon export and biogeochemical cycling and have been shown to contribute substantially to biomass in the North Atlantic, with some species especially dominant in subpolar regions during winter months (Bolaños et al., 2020). Robicheau et al. (2022) have shown that specific phytoplankton types have higher abundance values, especially in the early winter in the Bedford Basin when there are temperature anomalies. Primary production was not measured in this study, but modelling efforts have shown that the NPP in the Bedford Basin is in the 100's of mmol C m<sup>-2</sup> d<sup>-1</sup> at peak times, and 10's of mmol C m<sup>-2</sup> d<sup>-1</sup> otherwise (Black et al., 2023), suggesting that lower NPP in winter months can still yield modest POC fluxes. This was also observed in the nearby Gulf of Maine (Charette et al., 2001) where POC fluxes of  $14 \pm 0.8$  mmol C m<sup>-2</sup> d<sup>-1</sup> were measured in the lower range of their NPP measurements (26.4 mmol C m<sup>-2</sup> d<sup>-1</sup>). These small cells could possibly be the driver of this distinct winter flux (most prominent in the small particle fraction) that we see in January 2022, though the reasoning behind the winter flux remains ambiguous.

These results highlight the dynamic particle processes in Bedford Basin that maintain persistent <sup>234</sup>Th deficits, emphasizing the need for continued year-round, high-resolution particle and radionuclide measurements.



*Figure 9:* Profiles of the total residual activity of particulate  $^{234}$ Th (Ra<sub>P234</sub>) at the Compass Station vs normalized depth calculated as sample depth divided by total water depth (0 = surface, 1 = bottom). Ra<sub>P234</sub> values represent the sum of residual counts from both size fractions. Only Ra<sub>P234</sub> signals above the detection limit were found on the small particulate fraction. The dashed vertical line is the detection limit (derived from filter blank analyses).

#### 4.3. Non-steady state (NSS) & lateral heterogeneity considerations

Steady-state (SS) dynamics are often assumed in coastal and open ocean <sup>234</sup>Th/<sup>238</sup>U disequilibrium studies (e.g. Coale and Bruland, 1985; Foster and Shimmield, 2002) due to sampling constraints. Fluxes in coastal zones can however be impacted by lateral advection and non-steady state conditions, though the absolute value of each of these impacts on the total flux of <sup>234</sup>Th is difficult to evaluate without greater spatial and temporal resolution of <sup>234</sup>Th sampling. Thus, the SS model is applied here, and we explore the implications of NSS factors in this section.

In this study, we assume that  $\partial^{234}$ Th/ $\partial t$  is equal to 0 and the system is in steady state. The SS model assumes that i)  $^{234}$ Th activities and removal rates are near constant with respect to <sup>234</sup>Th decay, ii) the contributions of vertical and horizontal advection of <sup>234</sup>Th are minor, and iii) scavenging is irreversible. Physical studies of the Bedford Basin have generally shown that 1D assumptions hold for most of the year (Burt et al., 2013), and modelling efforts have demonstrated that ~ 80 % of particles that originate in the basin stay in the basin (Shan and Sheng, 2012). The circulation pattern of Halifax Harbour is characterized as a two-layer, estuarine-type system, with a fresher upper layer flowing seaward, driven by inflows from the Sackville River, and a saltier deep-water return flow (Fader and Miller, 2008). However, with a mean surface outflow of 0.2 cm s<sup>-1</sup>, this circulation is weakest in the Bedford Basin (Burt et al., 2013). Despite an inflow from the open Atlantic, the presence of a sill largely prevents mixing below 20 m in the basin (Shan et al., 2011). Although our observations come from a single, centrally located station, we acknowledge that lateral heterogeneity and edge effects may influence 234Th and POC dynamics in the Bedford Basin, especially from 0-20 m. To assess this, we evaluated a surface lateral <sup>234</sup>Th contribution using our 0-20 m mean <sup>234</sup>Th activity, a 3 km cross-basin scale, and observed basin current velocities (0.2 cm s<sup>-1</sup>). For a theoretical 50 % cross basin difference in  $^{234}$ Th activities, the lateral  $^{234}$ Th flux contribution is  $\sim 346$  dpm m $^{-2}$  d $^{-1}$ , which is  $\sim 52$  % of our mean surface cumulative flux, and ~ 14 % of the full water column flux to 65 m. This suggests that even under a strong 50 % contrast, the full water column budget changes only modestly. These lateral <sup>234</sup>Th flux estimates are intended as a first order, illustrative calculation, and real cross-basin gradients <sup>234</sup>Th activities (if any) should be measured for a more robust estimate of lateral

NSS impacts (rapid temporal changes in  $^{234}$ Th activities) have been shown to play a role on flux estimates during phytoplankton blooms, and prior studies have shown that NSS terms during blooms can alter  $^{234}$ Th export fluxes often by 10–50% (Buesseler et al., 1992; Ceballos-Romero et al., 2018). To assess how these departures from SS may impact our reported fluxes, we applied theoretical NSS sensitivities to our spring profiles. Assuming a modest NSS term (0.003 dpm L $^{-1}$  d $^{-1}$ ), the NSS correction is 7.8% of our reported SS  $^{234}$ Th flux. Under an upper bound scenario with a larger NSS term (0.01 dpm L $^{-1}$  d $^{-1}$ ), the resulting NSS correction increases to  $\sim 25.8$ % of the spring SS estimate. These theoretical corrections show that though minor, NSS effects can become significant during rapid bloom events. Periodic deep vertical mixing, such as during an intrusion event could increase the potential to disrupt the validity of the SS model. Indeed, we do see some indication of mixing multiple times throughout the study period (Fig. 2e), however, no major lines of evidence in the  $^{234}$ Th activities are indicated in our study (no major Ra $^{234}$  signals, or significant difference in POC: $^{234}$ Th), and despite intrusion events, the  $^{234}$ Th<sub>tot</sub> activities are similar between January 2024 (intrusion) and January 2022 (no intrusion). These observations may suggest that the Bedford Basin rapidly re-establishes a  $^{234}$ Th deficit following intrusion or mixing events and implies that particle scavenging and export are strong and continuous. If feasible, future studies in the basin should sample on a finer temporal resolution (1 $^{-4}$  weeks) to better understand NSS contributions in the basin.

# 4.4. Revisiting $^{234}$ Th activities on $> 51 \mu m$ particles









contributions in future studies.

A key source of uncertainty in estimating POC fluxes using <sup>234</sup>Th/<sup>238</sup>U disequilibrium is the POC:<sup>234</sup>Th ratio used for filtered particles, which has led to inconsistent flux estimates in previous studies (Lalande et al., 2008). Determining the most representative POC:<sup>234</sup>Th ratio remains an active area of research (e.g. Buesseler et al., 2006; Puigcorbe et al., 2020) to ensure accurate flux measurements. In the Bedford Basin, we find the primary source of variability in POC flux estimates is the POC:<sup>234</sup>Th ratio of large (> 51 μm) particles. Previously, Black et al. (2023) reported ratios in the Bedford Basin > 900 μmol

dpm<sup>-1</sup>, yielding exceptionally large fluxes (~ 2100 mmol C m<sup>-2</sup> d<sup>-1</sup>, Table 2), over an order of magnitude greater than predicted coastal carbon export models (Siegel et al., 2014). By contrast, this present study finds lower, more modest fluxes (< 70 mmol C m<sup>-2</sup> d<sup>-1</sup>), consistent with other coastal studies (< 200 mmol C m<sup>-2</sup> d<sup>-1</sup>, Luo et al., 2014; Cai et al., 2015; Cochran et al., 1995). High POC:<sup>234</sup>Th ratios (> 100 μmol dpm<sup>-1</sup>) have been previously linked to seasonal variability (Charette et al., 2001), however, the extreme values in Black et al. (2023) likely result from the differences in particle collection and delayed counting of beta emissions. Variations in sampling approaches highlight the potential for significant differences in POC measurements. Black et al. (2023) used a vacuum filtration system on Niskin-bottle collected samples, while this study employed McLane<sup>TM</sup> Large Volume Pumps that filter in-situ. In their comparative study (Graff et al., 2023) found that in-situ pump sampling resulted in substantially different POC concentrations compared to Niskin sampling at similar depths. This difference may arise due to variability in filtration speed, particle solubilization (Buesseler et al., 1995), and size fractionation (Hung et al., 2012). The methodological differences suggest that the elevated large particle POC:<sup>234</sup>Th ratios reported by Black et al. (2023) likely result from sampling artifacts rather than true variability in particle flux. The ratios obtained in this study provide a more representative measure of large particle POC:<sup>234</sup>Th at this site. To improve comparability, future studies in the Bedford Basin and other coastal regions should adopt consistent sampling methods to minimize methodological biases.

# 4.5. POC export in the Bedford Basin and an estimated C budget

The POC export fluxes derived in this study are in good agreement with other estimates of carbon fluxes taken in the Bedford Basin (Table 3). The mean depositional organic carbon flux (at 65 m), derived here from the large particle POC: $^{234}$ Th ratio and integrated  $^{234}$ Th flux was  $20 \pm 14$  mmol C m<sup>-2</sup> d<sup>-1</sup> (Fig. 7), two orders of magnitude lower than first estimates of  $^{234}$ Th derived POC flux (Black et al., 2023). This is a similar range as the nearby Gulf of Maine ( $^{34}$  ± 44 mmol C m<sup>-2</sup> d<sup>-1</sup>; March, June, September; Charette et al., 2001), Saanich Inlet in western Canada ( $^{11.5}$  ± 4 mmol C m<sup>-2</sup> d<sup>-1</sup>; Luo et al., 2014), Greenland continental shelf ( $^{25}$  ± 14 mmol C m<sup>-2</sup> d<sup>-1</sup>; Cochran et al., 1995), and Norwegian fjords ( $^{21.9}$ -24.4 mmol C m<sup>-2</sup> d<sup>-1</sup>; Wassmann, 1984). Our measured values are also similar to model estimates of organic carbon deposition flux in the Bedford Basin ( $^{25.2}$  mmol C m<sup>-2</sup> d<sup>-1</sup>; Rakshit et al., 2025), and sediment trap estimates at 60 m ( $^{17}$  mmol C m<sup>-2</sup> d<sup>-1</sup>; Hargrave and Taguchi., 1976). Translated to an annual molar estimate, (assuming our spring, summer, winter mean is representative of the year, and multiplying the mean daily flux by 365), our study reveals that  $^{7.3}$  ± 5.1 mol C m<sup>-2</sup> yr<sup>-1</sup> is delivered to the seafloor annually, similar to annual mean trap estimates in the Bedford Basin of 6.2 mol C m<sup>-2</sup> yr<sup>-1</sup> (Hargrave and Taguchi, 1976). The modeled daily benthic carbon flux in the Bedford Basin of 19 mmol C m<sup>-2</sup> d<sup>-1</sup> (Black et al., 2023), is equivalent to an annual export of  $^{\sim}$  7 mol C m<sup>-2</sup> yr<sup>-1</sup>, also consistent with our  $^{234}$ Th-based estimates.

Using the mean wt % POC in the upper 5 cm of the sediment cores (5.7 %), mean sedimentation rate (0.2 cm yr<sup>-1</sup>; Black et al., 2023) and mean dry bulk density (275 kg m<sup>-3</sup>), we find an organic carbon burial rate of 6.9 mmol C m<sup>-2</sup> d<sup>-1</sup>. This burial rate, combined with our mean POC depositional flux at 65 m, implies that the carbon burial efficiency is ~ 33%, larger than previous estimates at this site (10 %; Black et al., 2023), and similar to other coastal zones (Faust and Knies., 2019). The remaining ~ 67 % of the depositional POC is remineralized before burial, a finding consistent with measured dissolved inorganic carbon (DIC) fluxes from the sediments (15.6 mmol DIC m<sup>-2</sup> d<sup>-1</sup>; Rakshit et al., 2025). These findings suggest that the Bedford Basin is an important site of organic carbon sequestration. To better understand C dynamics in the ocean, carbon flux metrics have been developed that help define the efficiency of the Biological Carbon Pump in marine environments (Buesseler et al., 2020; Buesseler and Boyd 2009). The thorium export efficiency ratio (ThE-ratio) is the ratio of POC flux (based on <sup>234</sup>Th measurements) relative to NPP in the euphotic zone. To estimate the ThE-ratio in the Bedford Basin, we divide the annual mean

flux of POC measured in this study by the annual mean primary production in the basin. Primary production is rarely measured directly in the Bedford Basin with only one record of NPP from incubation studies (50 mmol C m<sup>-2</sup> d<sup>-1</sup>; Platt, 1975). Black et al. (2023) approximated an annual mean NPP of 66 mmol C m<sup>-2</sup> d<sup>-1</sup> (range 0.4-184 mmol C m<sup>-2</sup> d<sup>-1</sup>) using a model based on in-situ chl-a and PAR data, which falls within regional estimates from the nearby Gulf of Maine (26–399 mmol C m<sup>-2</sup> d<sup>-1</sup>; Charette et al., 2001). Using this NPP and our mean POC export at the base of the euphotic zone (~ 20 m; 29.4 mmol C m<sup>-2</sup> d<sup>-1</sup>), the export efficiency (ThE) is 44.5 %. Applying a ± 20 % sensitivity to NPP shifts ThE to ~ 37–56 %. This ThE ratio is comparable to other coastal sites (26%, Charette et al., 2001; 50%, Falkowski et al., 1988; 60%, Cochran et al., 1995). A ThE ratio of 44.5 % suggests moderate export efficiency at our site. It is, however, important to note that our estimation relies on modeled NPP values and updated in-situ measurements of NPP would yield a more precise value of these ThE estimates.

Table 3. An updated carbon budget of the Bedford Basin. Carbon fluxes derived in this study alongside comparable carbon flux measurements in the Bedford Basin. The measured fluxes reported in Black et al. (2023), which likely arise from their different particle sampling approach, are excluded here to enable a meaningful comparison.

| Carbon fluxes in the Bedford Basin         |                                        |                          |  |
|--------------------------------------------|----------------------------------------|--------------------------|--|
| Fluxes in the water column                 | mmol C m <sup>-2</sup> d <sup>-1</sup> | Methodology              |  |
| Large particle POC flux (60–65 m)          | 20 ± 14                                | POC: <sup>234</sup> Th   |  |
| Small particle POC flux (60–65 m)          | $12\pm7$                               | POC: <sup>234</sup> Th   |  |
| Total particulate flux <sup>a</sup> (60 m) | 17                                     | Cylindrical traps        |  |
| Modelled POC flux <sup>b</sup>             | 25.2                                   | Reaction-transport model |  |
| Modelled POC flux <sup>c</sup>             | 19                                     | Chl-a model              |  |
| Fluxes in the sediment                     | mmol C m <sup>-2</sup> d <sup>-1</sup> | Methodology              |  |
| Large particle sediment accumulation flux  | $22 \pm 9$                             | POC: <sup>234</sup> Th   |  |
| Small particle sediment accumulation flux  | $13 \pm 8$                             | POC: <sup>234</sup> Th   |  |
| Measured DIC flux <sup>b</sup>             | $15.6\pm10$                            | Pore-water extraction    |  |

<sup>&</sup>lt;sup>a</sup> Total particulate carbon flux from cylindrical trap estimates (Hargrave and Taguchi, 1978). Sediment traps did not have an efficiency correction and may not be limited to POC.

## 565 5. Conclusions



This study demonstrates the utility of the coupled <sup>234</sup>Th water column–surficial sediment sampling approach by quantifying <sup>234</sup>Th and POC fluxes in the water column and surface sediments of Bedford Basin, a coastal fjord in the North Atlantic, over quasiseasonal sampling campaigns from 2021 to 2024. Within the uncertainties of the <sup>234</sup>Th/<sup>238</sup>U disequilibrium method and a steady state approach, we find that the integrated deficit of <sup>234</sup>Th in the water column matches the excess of <sup>234</sup>Th in the uppermost sediments well. The match suggests that, on a timescale of the ~ 100 days, the Bedford Basin is in balance with respect to <sup>234</sup>Th, and no major particle loss is occurring. There were significant deficits of <sup>234</sup>Th relative to its parent <sup>238</sup>U throughout the water column during all sampling events, including boreal winter. We conclude that the majority of <sup>234</sup>Th<sub>tot</sub> in the Bedford Basin is associated with the particulate <sup>234</sup>Th pool, with a significant portion bound to the small (1–51 μm) particle

<sup>&</sup>lt;sup>b</sup> From Rakshit et al. (2025).

<sup>&</sup>lt;sup>c</sup> Model derived from Black et al. (2023) using chl-a measurements.

fraction. Translated into POC fluxes, the observed deficits reveal a mean POC export flux to the seafloor of 20 ± 14 mmol C m<sup>-2</sup> d<sup>-1</sup>. This estimate is based on the integrated <sup>234</sup>Th flux between (0–65 m water depth) multiplied by the POC:<sup>234</sup>Th ratio on large (> 51 μm) particles and is on par with other carbon flux proxies in the area. Water column POC flux estimates showed no apparent trend with season, other than a slightly elevated flux corresponding to elevated chl-a in May 2023. Measurements of POC export in boreal winter reveal modest flux, despite lower chl-a signals at this time. We estimate a thorium export ratio of 44.5 % and an organic carbon burial efficiency of 33 %, higher than previously estimated at this site. Based on this data, we suggest that the biological carbon pump is moderately efficient in the Bedford Basin, and that this site, like many other coastal areas, is an important site of organic carbon sequestration.

Data Availability: Data to be uploaded to <a href="https://www.bco-dmo.org/">https://www.bco-dmo.org/</a>

Author Contributions: MH: Conceptualization, Formal Analysis, Investigation, Methodology, Visualization, Writing – original draft, Writing – review & editing. EB: Conceptualization, Investigation, Writing – review & editing. CA: Conceptualization, Investigation, Methodology, Resources, Writing – review & editing. MA: Formal Analysis, Methodology, Writing – review & editing. SK: Conceptualization, Funding acquisition, Investigation, Supervision, Writing – Review & Editing. Competing Interests The contact author has declared that none of the authors have any competing interests.

Acknowledgements: This research was supported by the Ocean Frontier Institute (OFI), Natural Sciences and Engineering Research Council of Canada (NSERC), Nova Scotia Graduate Scholarship (NSGS). The authors would like to thank the crew of the Connor's Diving Ltd. vessel EASTCOM, Laura deGelleke, Richard Cheel, Subhadeep Rakshit, and Adam White for assistance in the field. We thank the Bedford Institute of Oceanography for maintaining the time series station at the Bedford Basin and making the data accessible. We thank Claire Normandeau (Dalhousie CERC.OCEAN) and Joshua Landis (Dartmouth College) for analytical assistance.

**Competing Interests:** The contact author has declared that none of the authors have any competiting interests.

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
