# Peer review of "Balancing water column and sedimentary 234Th fluxes to quantify coastal marine carbon export"

_EGUsphere, 2025_

## Referee Comment (RC1)

**Review of *Balancing water column and sedimentary $^{234}$Th fluxes to quantify coastal marine carbon export*, by Healey et al.**

This article provides quasi-seasonal coupled water column and surficial sediment sampling of Th-234 and POC to evaluate and refine the carbon budget in the Bedford Basin. I find the research necessary because, as highlighted by the authors, not many studies using the commonly applied approach of Th-234 to assess carbon export have been conducted in coastal areas. Additionally, the combination of water column and sediment measurements is not a usual approach either, yet it provides complementary data allowing to better understand the fate of the particle fluxes in coastal areas. By combining both types of sampling, the authors conclude that there is no major particle loss in the Beadford Basin and that the magnitude of the export and burial fluxes is similar to those reported in previous studies in coastal areas, which are relevant sites for organic carbon sequestration.

The paper is well presented, the text and the figures are clear and provide the necessary information to support their findings. I have specific comments regarding some aspects that need clarification or small corrections, but overall, the study is robust and contributes to the growing body of literature discussing carbon budgets in coastal environments.

Some specific comments are detailed below:

- **L111**: It mentions that the water samples were collected with 10L Nisking bottles and that, after Th-230 was added, 2L were processed for total Th-234. I believe the way it is written can lead to confusion, since it might look like 10L were collected, spiked and then 2L subsampled, which I believe was not the case.

- **L143**: Please specify the pore size used to obtain filtered and UV treated seawater (no need to specify in that line the pore size of QMAs since it is mentioned in L141-142)

- **L152-154**: "*For comparison, the POC:$^{234}$Th ratio on the small particle fraction was also calculated using Eq. 4*" is not correct. Please rephrase since Eq. 4 calculates POC flux, no ratios.

- **L151-154** and **Eq. 4** can be simplified by adding a subscript next to the ration in the formula so that the text just needs to say that POC fluxes are calculated using Eq. 4 using the ratios of the different particle fractions.

- **L157-165**: The first two sentences seem a bit repetitive and can be streamlined. I think the repetition of "*novel proxy and residual beta activity*" makes it look repetitive. Use the "abbreviation" $Ra_{p243}$ instead after defining it at the beginning.

- **L169**: Can you provide the uncertainty of $n_{p2}$ and the blank? These blank counts come from the dipped filters, correct? Also in that line, notice that cpm was used before and defined later.

- **Section 2.5** Excess $^{234}$Th inventories requires more detail:

  o How thick were the sections of the cores (0.5 cm, 1 cm) and how far down where they cut/measured ( 6 cm, based on Fig. 8?)

  o How quickly were the cores cut after collection? Were they kept in a freezer until processing?

  o How were they dried? For how long approx.? Was the weight checked multiple times to ensure that the sediment was dry?

  o Was the dry weight corrected for salt content?

  o How much sediment was approximately used for the measurement?

  o How long were the samples measured for and how long after collection (since Th-234 has a 24.1 d half-life)?

  o How was the mean supported activity obtained? Did you remeasure all the sections or just certain ones? L465 suggests only section 4.5 to 6 was used as supported. Please provide more details.

  o What type of gamma detector was used? The gamma counting was done using the petri-dish, so it was not done in a well-type gamma counter, right? Mention it.

- **L191-192**: Please explain how the organic carbon samples were processed before measurements. Were they also fumed with HCl, as per filters?

- **L205**: Please provide a threshold of Chl-a or some definition for the blooms in the region or a usual range of Chl-a values during blooms. The text reads that a 11.1 mg/m$^3$ chl-a signal is a weak bloom but would be good to have some sort of a scale to assess that.

- **Figure 4**: "*Total particulate $^{234}$Th is equal to the sum of the large and small particles*" Why is that sentence there for Th-234 and not for POC? Shouldn't the y-axes just be Particulate POC and Particulate $^{234}$Th, since there are different size fractions represented in both plots?

- **L431**: "*POC export fluxes (Fig. 6) were calculated following Eq. 4 and using >51um ratios (Fig. 5a)*" it's correct, but not complete, because the export fluxes shown in Fig. 6 are also calculated using the 1-51um ratios from Fig. 5b. Please refine this paragraph to streamline it. You could cite both figures and then start talking about the small fraction fluxes, were no patter with depth was observed, and then talk about the large fraction fluxes, since they are discussed a bit more.

- **Figure 6**: The a) should be moved in front of "*Profiles of cumulative….*" since "*Bedford Basin 1D steady state fluxes in 2021-2024*" applies to the three panels. Make the y-axes to match between the three panels, otherwise it creates a visual distortion as if the POC ones were deeper. Why all the different symbols in the Th-234 flux? Pick one, as done for the POC fluxes based on particle size. Also, the R$^2$ shown in graphic a) is not explained. Maybe add the whole linear regression and cite the figure in the main text (L428).

- I am a bit confused regarding the information provided in Fig. 6 in the sediment part of panels b) and c) and what it is shown in Fig. 7. Isn't it the same data? Why show it twice? I would combine these figures as if Fig. 7 was a zoom-in of the sediment part of Fig. 6b and Fig. 6c. Please double check that the numbers match, I have the impression that,

for example, Jan_2024 >51um POC flux shown in Fig. 6b is higher than when plotted in Fig. 7a

- **L462-463**: It is confusing to say that the EQ depth was always ~4.5cm and then write that the majority of $Th_{xs,0}$ was confined to the top ~2-4.5cm. Also, looking at Figure 8 (please cite figure in that first sentence), the largest excess is found in the upper 1.5 cm.

- **L463**: "*Excess $^{234}Th$ in the EQ was variable*". There should not be excess $^{234}$Th at the equilibrium depth. Do you mean above the EQ?

- **L467**: "*were consistent with core depth between 0 – 4 cm*". Not sure what do you mean. Are you referring to the concentrations and percentages found in those depths in other similar cores? Or do you mean that concentrations did not vary in the upper 4 cm of this study's cores? What concentrations and percentages of POC did you find between 4 cm and 6 cm? Was POC only measured in the upper 4 cm? There are POC:Th ratios deeper than 4 cm, were they extrapolated?

- **Figure 8**: Please polish the caption for panel b). The a) should be placed before "decay corrected". b) says "*POC:$^{234}Th_{xs,0}$*" but the title says "*POC:$^{234}Th$*". Also, no need to explain the supported in the caption, it has been explained in the main text and it is not shown in the figure.

- **L559-560**: Could you expand on Lampitt (1985) statement? I understand the previous sentence about the fact that the ratios would be higher if there was resuspension of the top sediments, but I feel like Lampitt's sentence is not properly integrated in the discussion or lacking some more information. Not only the composition of the particles in the sediment-water interface facilitates resuspension, but hydrodynamics are also crucial, meaning that, even if those particles are lithogenic and not detrital, with enough hydrodynamics they can be resuspended. Lampitt's 1985 study area is also 4000 m instead of the 70 m at Compass Station. Not sure it is the best comparison to make, unless you want to explain a bit more.

- **L567-569**: Boetius et al. 2013 talks about large algal chunks found at depth fresh and seen by cameras, not sure they refer to measuring fluorescence along the water column and finding high values at depth, so I am not sure the comparison applies. In any case, I would assume the deeper fluorescence signals found in this study are due to mixing, having some surface phytoplankton reaching deeper layers. Yet, blooms take place when water column presents higher stratification. Did you do isotopic analyses on the POC samples?

- **L567-573**: A high particle load is mentioned. Is that based on visual assessment (the water looked turbid) or there was some kind of measurement done that would indicate that (e.g., light penetration).

- **L580-582**: Bolanos et al. (2020) citation refers to importance of small phytoplankton in terms of biomass. Mention it.

- **L583-584**: Cite the reference for the *Synechococcus* statements.

- **Fig. 9**: Please clarify y-axis, I don't think I understand it and it is not described in the caption.

- **L605**: Just to clarify, the 8 dots that are above the detection limit had 0 $RA_{P234}$ from the large fraction, so those values are entirely from the small fraction particles or the

difference from the detection limit value and the $RA_{P234}$ value in those cases is coming entirely from the small particles?

- **Section 4.4** I appreciate the authors revisiting Black et al (2023) POC:Th data and discussing the discrepancies observed. They could have avoided it and focus only on their current dataset, but instead they highlight the substantial differences and mention the potential sampling artifacts causing the extremely elevated ratios in that previous work.

- **L648**: Table 3 does not show other biogeochemical measurements, it shows estimates of POC export fluxes by others using other methods.

- **L654**: Also similar to modelled POC flux reported in Black et al. 2023, based on the data shown in Table 3.

- **L662-663**: I obtain slightly different burial rate, which leads to slightly different burial efficiency. Minimal differences, but please double check the numbers.

- **L671**: Add at the end of the sentence "i.e., export efficiency". The following sentence starts talking about export efficiency and not ThE-ratio and might seem like they are different things for those not familiar with the thorium approach.

- **L672-673**: It is the other way around, export/production = ThE; as stated in L676: "*divide the mean POC... by the mean NPP*"

- **L677**: 29.4 mmol C $m^{-2}$ $d^{-1}$ / 66 mmol C $m^{-2}$ $d^{-1}$ = 44.5% not 49% Please double check the numbers. L679 also mentions 49% ThE ratio and in the conclusions. Also check the burial efficiency reported in the conclusions (see my comment for L662-663).

- **Table 3**:

  o The value reported by Hargrave and Taguchi (1978) is under "*Particulate flux*" and the units are in mmol C $m^{-2}$ $d^{-1}$. If it is not POC flux (otherwise I assume it would have said it) it should be total C to match the units reported. Is that the case? Please specify it in the title: "total C flux" or provide the necessary information as a table foot note.

  o The fluxes in the sediment reported in Black et al. 2023 are not shown in this table due to their extremely high values because of the particle sampling method used there, correct? It would be good to expand the first sentence of section 4.5 to explain the summary table 3, what was included in it and what not, and why.

Technical comments:

- **L124 Eq. 1** and **L126 Eq 2**.: Make sure the superscripts are shown as such for $^{238}$U and $^{234}$Th

- **L166**: Eq. 5 uses $Ra_{p243}$ but later on, in L603-605 $RA_{P243}$. Check, this term is used in other lines not cited here.

- **L190**: Multiplied by

- **L191**: Add (OC) after "Organic carbon" since it is used later in L194, or write it entirely in L194.

- **Figure 2 panel E**: The image definition of this panel is worse than the others. Also, please provide the y-axes titles in the corresponding side of the graph. The legend is split but not the actual titles. The vertical line on the right side of the panel looks odd. Can it be just the actual y-axis with its values? Also, the font size of the y-axes is much smaller than for the other panels. L281, T, S, O have not been defined before, add them in the previous line.

- **Figure 2**: Add depth [m] also in panels B, C and D.

- **Table 1**: Nowhere in the text nor in the caption it is defined what LSF or SSF stands for. One can figure it out, but it would be good to add the whole wording in the caption.

- **L404**: please cite the number of data point considered (p < 0.01, n= X)

- **Figure 5**: The legend is the same for both panels, even though the shape of the symbols is different for large and small particles, correct? The same symbol could be used while highlighting the size fraction a bit more (maybe add it to the title?).

- **L425**: "*spring summer 2019*" missing an "and"

- **L436**: BB for Bedford Basin is only used here and in L657-658. I would just right the whole name in those lines.

- **L436**: "*...that this event brought additional...*" remove "it"

- **L474** cites only Fig.6b and 6c but it could also cite Fig. 7. Although, see my previous comment regarding the combination of these two figures in one.

- **Figure 8**. I don't mind the different symbols for the different sampling periods (although this is not done in the previous figures) but they seem to have different size? It could be an optical effect due to the different shape/color, but the dark blue triangles seem bigger than the rest.

- **L511**: $^{234}$Th appears as $^{234Th}$

- **L590**: It says "Jan 2022" but in all other instances the whole months it is written down, i.e., January. Same in L619 and L620.

- **L603-605**: Caption of Figure 9 – the residual activity here it is expressed as RA$_{P234}$, different than in L166. Check the whole text and make sure to be consistent.

---

## Author Comment (AC1)

The study by Healey et al., investigates the water column $^{234}$Th – $^{238}$U disequilibrium and surficial sediment inventory of excess $^{234}$Th to constrain the particulate organic carbon (POC) export and burial in Bedford Basin. The integration of water column and sediment data to quantify the POC fluxes is appreciable.

Authors conducted quasi-season sampling from 2021 – 2024, capturing the seasonal dynamics over three-year period. The water column and sedimentary $^{234}$Th fluxes and POC fluxes along with bottom sediment depositional POC fluxes are in close agreement which verifies the estimates of POC fluxes in water column and mostly contributed by the vertical setting of particles. The study is comprehensive with proper methodology.
* * *
We thank reviewer 2 for their time and very helpful comments. Below, we answer each of the reviewer's comments. The reviewer's comments are in blue; our responses are in black. We agree with the reviewer's concerns and have addressed them in a revised manuscript. Overall, the reviewer's comments have made the manuscript much stronger, especially the discussion of potential NSS effects.

- Though the Bedford Basin is relatively small, the data from a single central station might not capture the lateral heterogeneity or edge effects. In the center, as shown by the data, the POC contribution is mostly from the primary production or vertical setting of particles, however, the lateral contribution of the particles cannot be ignored, especially in this shallow fjord. It can be discussed in the manuscript in general regarding the overall basin for wider implications.

Response: Indeed, our data are from a single, centrally located station in the Bedford Basin and we agree with the reviewer that lateral heterogeneity could potentially contribute to $^{234}$Th and POC dynamics in this shallow fjord system. However, we anticipate that lateral impacts are small, as explained below. In the revised manuscript, we have added a statement in the discussion (section 4) that (a) acknowledges the limitation, (b) assesses its potential magnitude (see below) and (c) emphasizes that spatially resolved sampling across the Basin would be valuable in future studies. We propose that future efforts sample a transect from the outer harbour, and/or edges of the Bedford Basin to the central Compass Station in order to fully capture spatial heterogeneity and better assess the contribution of lateral inputs on both $^{234}$Th and POC flux.

The Bedford Basin is connected to the Halifax harbour & open Atlantic, and therefore one might expect some addition of total $^{234}$Th from these waters into the Bedford Basin. Although lateral impacts are not directly measured in this study, we anticipate that they are small for the following reasons:

Modelling efforts (Shan and Sheng, 2012) used particle tracking in the Bedford Basin and Halifax Harbour to demonstrate that ~ 80% of the particles that originate in the basin stay in the basin. The model implies that the observations in the central basin predominantly reflect vertical production and settling. Lateral exchanges occur only episodically during storms and strong tidal phases, but do not constitute a persistent flux pathway. Additionally, Shan et al. (2011) find that the sill

between the Bedford Basin and the outer harbour largely prevents mixing below 20 – 30 m water depth. We performed a simple first-order estimate of lateral advection using observed basin current velocities (surface ~ 0.2 cm s$^{-1}$, subsurface <1 cm s$^{-1}$; Shan and Sheng, 2012; Burt et al., 2013), a 1 km basin scale, and small cross-basin gradients in $^{234}$Th (0.01–0.03 dpm L$^{-1}$). This analysis suggests that lateral fluxes contribute typically < 10%, and maximally 20%, of the vertical flux depending on layer and assumed gradient. Additionally, based on numerical passive tracer experiments, Shan and Sheng (2012) estimate the e-folding flushing time of 40 – 90 days in the upper and entire Bedford Basin respectively. In our study, we find the residence time of particulate $^{234}$Th to be 18 ± 7 days, which is substantially shorter than, or at best comparable to, the advective flushing timescale. Thus, while episodic lateral exchanges may occur during storms or intrusions, multiple lines of evidence suggest that vertical production and settling dominate $^{234}$Th-POC dynamics in the Bedford Basin.

In our revised text, we explain in more detail the points discussed here and highlight lateral heterogeneity and edge effects in our discussion.

The hydrological parameters of the basin show similar patterns for different years and seasonal variations. The mixing and stratification are seasonal and the importance of absence of winter mixing and spring blooms in relation with steady state 1-D Model needs to be further discussed. The limited direct measurements of nepheloid layers and primary production weakened mechanistic interpretation of persistent $^{234}$Th deficits. Although 1-D steady state model is assumed in this study, the system shows several seasonal dynamic features, especially intrusion and spring bloom. The authors have included a paragraph of discussion about the NSS contribution for < 3%, but it doesn't justify this datasheet. The authors should describe how the absence of NSS might impact the fluxes especially during different seasons and spring bloom for this study.

Response: We thank the reviewer for this important comment. We agree that the Bedford Basin is a seasonally dynamic system with winter mixing, spring blooms, and episodic intrusions, all of which can potentially influence particle fluxes and $^{234}$Th distributions. Like many $^{234}$Th studies, our study used steady state assumptions due to the logistical constraints associated with field sampling. While we recognize that NSS conditions likely occur during certain periods, the application of the 1-D steady-state model allows for consistency and comparability with previous studies. In a revised version of the manuscript, we have expanded the discussion of NSS as follows:

- In section 4.3, we have expanded our discussion to address how our steady-state fluxes could be impacted by spring bloom and intrusion events.
- We apply a theoretical non-steady state sensitivity to our two spring profiles. Assuming a non-steady state term of 0.003 dpm L$^{-1}$ d$^{-1}$, the NSS correction is 7.8% of our reported SS $^{234}$Th flux. To also include an upper boundary spring bloom scenario, we also tested a larger non-steady state term of 0.01 dpm L$^{-1}$ d$^{-1}$. The resulting NSS correction is ~ 25.8% of the spring SS estimate we report. We also quantify and outline in 4.3 how these potential NSS contributions change our POC flux estimates.
- We briefly discuss and estimate the possible NSS contribution from intrusion events.

In conclusion, though we did not directly measure NSS impacts, we now report theoretical NSS fluxes in the revised manuscript and expand on their implications in the Bedford Basin. We note that repeat sampling (every 1–4 weeks) remains the preferred way to quantify NSS explicitly when feasible. We suggest that future work in the Bedford Basin should consider following a NSS sampling scheme to better constrain $^{234}$Th derived fluxes. Additionally, use of other shorter-lived tracers, such as Bismuth-210 ($t_{1/2}$ = 5 d) can capture even finer resolutions (hours-days).

I appreciate the authors for nice graphs and figures; however, the authors should include the $^{234}$Th profile data along with errors even if it's in supplementary data.

The $^{234}$Th profile data with errors are shown in Figure 3. However, the errors are small , so they might have been mistaken for symbols by the reviewer.

Fig 6 a. There is no need to include the $R^2$ value as this is cumulative fluxes and $R^2$ values doesn't have any meaningful essence.

We agree with the reviewer that the $R^2$ value does not provide additional scientific meaning in this context, as cumulative flux inherently increases with depth. We have removed the $R^2$ value in the figure.

Overall, the manuscript is valuable, relevant and methodologically sound. Incorporating these suggestions, addressing spatial heterogeneity would provide clarity and enhance the impact of this manuscript.

**Works cited:**

Burt, W. J., Thomas, H., Fennel, K., & Horne, E. (2013). Sediment-water column fluxes of carbon, oxygen and nutrients in Bedford Basin, Nova Scotia, inferred from $^{224}$Ra measurements. *Biogeosciences*, *10*(1), 53–66. https://doi.org/10.5194/bg-10-53-2013

Shan, S., & Sheng, J. (2012). Examination of circulation, flushing time and dispersion in Halifax Harbour of Nova Scotia. *Water Quality Research Journal*, *47*(3–4), 353–374. https://doi.org/10.2166/wqrjc.2012.041

Shan, S., Sheng, J., Thompson, K. R., & Greenberg, D. A. (2011). Simulating the three-dimensional circulation and hydrography of Halifax Harbour using a multi-nested coastal ocean circulation model. *Ocean Dynamics*, *61*(7), 951–976. https://doi.org/10.1007/s10236-011-0398-3

---

## Author Comment (AC2)

Review of Balancing water column and sedimentary $^{234}$Th fluxes to quantify coastal marine carbon export, by Healey et al.

This article provides quasi-seasonal coupled water column and surficial sediment sampling of Th-234 and POC to evaluate and refine the carbon budget in the Bedford Basin. I find the research necessary because, as highlighted by the authors, not many studies using the commonly applied approach of Th-234 to assess carbon export have been conducted in coastal areas. Additionally, the combination of water column and sediment measurements is not a usual approach either, yet it provides complementary data allowing to better understand the fate of the particle fluxes in coastal areas. By combining both types of sampling, the authors conclude that there is no major particle loss in the Bedford Basin and that the magnitude of the export and burial fluxes is similar to those reported in previous studies in coastal areas, which are relevant sites for organic carbon sequestration.

The paper is well presented, the text and the figures are clear and provide the necessary information to support their findings. I have specific comments regarding some aspects that need clarification or small corrections, but overall, the study is robust and contributes to the growing body of literature discussing carbon budgets in coastal environments.
* * *
We thank reviewer 1 for their thoughtful and detailed comments, which helped us to significantly improve the manuscript. We agree with all their comments and made the corresponding changes in the manuscript. Below, we answer each of their comments in detail. The reviewer's comments are in blue, our responses are in black.
* * *
- **L111**: It mentions that the water samples were collected with 10L Niskin bottles and that, after Th-230 was added, 2L were processed for total Th-234. I believe the way it is written can lead to confusion, since it might look like 10L were collected, spiked and then 2L subsampled, which I believe was not the case.

Response: In the revised manuscript, we clarify that 10 L Niskin bottles were used for sample collection, but only a 2 L subsample was spiked and processed for total $^{234}$Th measurements.

- **L143**: Please specify the pore size used to obtain filtered and UV treated seawater (no need to specify in that line the pore size of QMAs since it is mentioned in L141-142)

Response: The pore size used for seawater filtering was 0.2 μm. This is now mentioned in L143 of the revised manuscript. We have also removed the repetition regarding the QMA pore size,

which the reviewer correctly identified.

**L152-154**: "*For comparison, the POC:$^{234}$Th ratio on the small particle fraction was also calculated using Eq. 4*" is not correct. Please rephrase since Eq. 4 calculates POC flux, no ratios.

Response: We agree with reviewer 1 and have rephrased the relevant sentences. The POC flux can be calculated with either (a) the POC:$^{234}$Th ratio on small particles or (b) the POC:$^{234}$Th ratio on large particles using Eq. 4. We clarify in the manuscript that the fluxes reported using either a) or b) represent two alternative approaches to estimating flux, and that these should not be considered additive, but rather complimentary perspectives on export.

- **L151-154** and **Eq. 4** can be simplified by adding a subscript next to the ration in the formula so that the text just needs to say that POC fluxes are calculated using Eq. 4 using the ratios of the different particle fractions.

Response: Change made as suggested.

- **L157-165**: The first two sentences seem a bit repetitive and can be streamlined. I think the repetition of "*novel proxy and residual beta activity*" makes it look repetitive. Use the "abbreviation" Rap243 instead after defining it at the beginning.

Response: Change made as suggested.

- **L169**: Can you provide the uncertainty of np2 and the blank? These blank counts come from the dipped filters, correct? Also in that line, notice that cpm was used before and defined later.

Response: We thank reviewer 1 for pointing out these issues. As the reviewer correctly suggests, the blank counts ($n_0$) were obtained from dipped filters. Dipped filters are deployed with the pump to depth, but, importantly, no seawater is pumped through them. They can be considered a seawater process blank and are often referred to as a "dipped blank", following Lam et al., 2015.

We now report the uncertainties for both $n_{P2}$ and $n_0$ based on counting statistics.

The revised sentence reads:

"Where $n_{P2}$ is the second β counting rate of particulate $^{234}$Th ($0.30 \pm 0.07$ cpm), $n_0$ is the blank mean obtained from dipped filters ($0.24 \pm 0.02$ cpm), $\epsilon$ is the detector efficiency (0.3), and V is the volume of seawater pumped (L). The term "cpm" (counts per minutes) is now defined at its first occurrence.

- **Section 2.5** Excess $^{234}$Th inventories requires more detail:

Response: We have now revised section 2.5 in the manuscript to include more details on the method & processing of the sediments. The details are explained below.

o   How thick were the sections of the cores (0.5 cm, 1 cm) and how far down where they cut/measured (6 cm, based on Fig. 8?)

Response: Cores were generally sliced into 0.5 cm thick sections. In some upper core intervals, we took 1 cm thick sections, as the sediment was too soft to cut thinner sections. To clarify, all cores were measured for $^{234}$Th down to 10 cm, some cores were only measured down to 6 cm for TOC.

Ideally, we wanted to slice cores into 0.5 cm samples for higher resolution in the upper most sediments. If this was not possible, the cores were sectioned into 1 cm samples.

o   How quickly were the cores cut after collection? Were they kept in a freezer until processing?

Response: Cores were either sliced on the same day they were retrieved in the field or stored in a walk-in fridge (4°C) for 1 – 2 days (max) prior to slicing.

o   How were they dried? For how long approx.? Was the weight checked multiple times to ensure that the sediment was dry?

Response: The sediments were dried at 55◦C in our laboratory oven for approximately 24 hours. After this period, they were weighed and returned to oven and reweighed at regular intervals until a constant weight was achieved.

Response: Yes, the dry weights were corrected for salt content. Salt contribution from porewater was estimated based on salinity measurements of the overlying water and subtracted from the measured dry weight. This clarification has been added to the method section of the revised manuscript.

o   How much sediment was approximately used for the measurement?

Response: Approximately 15 g of dry sediment was used for gamma counting.

o   How long were the samples measured for and how long after collection (since Th- 234 has a 24.1 d half-life)?

Response: Gamma analysis was typically within less than 24 days, i.e., one half-life of sediment retrieval in the field. Samples were counted for 7 to 24 h depending on the time to achieve counting uncertainties of less than 5 % on the primary peaks.

o   How was the mean supported activity obtained? Did you remeasure all the sections or just certain ones? L465 suggests only section 4.5 to 6 was used as supported. Please provide more details.

Response: We thank reviewer 1 for this comment. All cores were measured for $^{234}$Th down to 10 cm. The mean supported $^{234}$Th activity was determined from the deepest sediment intervals in each core where no excess $^{234}$Th was detectable, i.e., below the penetration depth of excess $^{234}$Th. For our cores, this was consistently at ~4.5–6 cm depth (i.e. at 4.5–6 cm, the $^{234}$Th activities were consistent and showed no further decrease in $^{234}$Th activity). These deepest sections were measured directly and used to define the supported activity, which was

then subtracted from the total $^{234}$Th activities in the overlying sediments to calculate excess $^{234}$Th. We have now clarified this in the methods section.

- o *What type of gamma detector was used? The gamma counting was done using the petri-dish, so it was not done in a well-type gamma counter, right? Mention it.*

Response: Correct. Gamma counting was conducted using Canberra Intrinsic High-Purity Germanium (HPGe) detectors in a planar configuration, not a well-type of detector. Samples were counted in low-background geometry using petri dishes, which were placed directly on the detector surface to optimize efficiency for low-energy gamma emissions (such as the 63.3 keV peak of $^{234}$Th). We have added this information in the revised paper.

- **L191-192**: *Please explain how the organic carbon samples were processed before measurements. Were they also fumed with HCl, as per filters?*

Response: We did not fume the sediment samples with HCl prior to CN analysis, and spell this out in the revised text.

Previous work on Bedford Basin sediments has compared acid-fumigated and non-acid-fumigated sediment samples (Black et al., 2023). The average carbon content did not differ between the acid fumigated ($5.5\% \pm 0.5\%$, n=29) and non-acid fumigated ($5.7\% \pm 0.8\%$, n=39). This published finding is consistent with additional unpublished measurements that were made over years at Dalhousie University. Collectively, the data imply that most of the sedimentary carbon is organic in nature, which is why fumigation is not necessary. We thank the reviewer for their comment, which allowed us to clarify the text.

- **L205**: *Please provide a threshold of Chl-a or some definition for the blooms in the region or a usual range of Chl-a values during blooms. The text reads that a 11.1 mg/m$^3$ chl-a signal is a weak bloom but would be good to have some sort of a scale to assess that.*

Response: Based on observations made by the Bedford Basin Monitoring Program since 1993, "background" chlorophyll-a typically ranges from ~3 to 10 mg m$^{-3}$ (surface, 1–10 m) in the summer. In contrast, spring blooms range from ~7 to 10 mg m$^{-3}$ and up to ~28 mg m$^{-3}$. Extreme short-lived events have reached chlorophyll concentrations over 100 mg m$^{-3}$. Therefore, we define a "weak bloom" in Bedford Basin as chlorophyll-a concentrations just above the summer background (e.g., >10 mg m$^{-3}$) but below intense bloom threshold of 28 mg m$^{-3}$. We will include these ranges and citations to make the terminology clearer in a revised manuscript.

- **Figure 4**: *"Total particulate $^{234}$Th is equal to the sum of the large and small particles" Why is that sentence there for Th-234 and not for POC? Shouldn't the y-axes just be Particulate POC and Particulate $^{234}$Th, since there are different size fractions represented in both plots?*

Response: We agree that axis labels and definitions should reflect the size-fractionation. In our study, particulate material was collected sequentially on a 51 μm screen and a 1 μm filter from the same pump casts. After corrections for blanks, yield (for $^{234}$Th), and volume,

we define total particulate $^{234}$Th (>1 µm) as the sum of $^{234}$Th (1–51 µm) and $^{234}$Th (>51 µm). Similarly, total particulate POC (>1 µm) is defined as the sum of POC (1–51 µm) and POC (> 51 µm). We have clarified this in the methods section of the revised manuscript and updated the figure axes to read "Particulate $^{234}$Th (>1 µm)" and "Particulate POC (> 1 µm)". The legend now also correctly identifies the 1–51 µm and >51 µm fractions.

- **L431**: "*POC export fluxes (Fig. 6) were calculated following Eq. 4 and using >51um ratios (Fig. 5*a)" it's correct, but not complete, because the export fluxes shown in Fig. 6 are also calculated using the 1-51um ratios from Fig. 5b. Please refine this paragraph to streamline it. You could cite both figures and then start talking about the small fraction fluxes, were no pattern with depth was observed, and then talk about the large fraction fluxes, since they are discussed a bit more.

Response: We thank the reviewer for this suggestion. We have revised the paragraph to clarify that fluxes were calculated using both the small (1–51 µm) and large (>51 µm) POC:$^{234}$Th ratios (Figs. 5b and 5a, respectively). We now introduce both fractions, then describe the small fraction first, followed by the large fraction, where more variability was observed. L431 now reads "POC export fluxes (Fig. 6) were calculated following Eq. 4 using both the 1–51 µm and >51 µm POC:$^{234}$Th ratios (Figs. 5b and 5a, respectively). Fluxes based on the small size fraction (1–51 µm) showed no consistent pattern with depth, while fluxes from the large size fraction (>51 µm) varied more strongly. At the deepest depth sampled (60–65 m), large fraction POC fluxes had a mean (± s.d.) of 20 ± 14 mmol C m$^{-2}$ d$^{-1}$ (Fig. 7a) and ranged from 3.6 ± 1.1 to 44.5 ± 4.1 mmol C m$^{-2}$ d$^{-1}$."

- **Figure 6**: The a) should be moved in front of "*Profiles of cumulative….*" since "*Bedford Basin 1D steady state fluxes in 2021-2024*" applies to the three panels. Make the y-axes to match between the three panels, otherwise it creates a visual distortion as if the POC ones were deeper. Why all the different symbols in the Th-234 flux? Pick one, as done for the POC fluxes based on particle size. Also, the R$^2$ shown in graphic a) is not explained. Maybe add the whole linear regression and cite the figure in the main text (L428).

Response: We thank the reviewer for their insightful comments and keen eye. We have since made the appropriate changes by making the y-axis consistent and choosing one symbol. The R squared value comes from the linear regression of all dates (L427). This R-squared value will be added into the text of the manuscript.

I am a bit confused regarding the information provided in Fig. 6 in the sediment part of panels b) and c) and what it is shown in Fig. 7. Isn't it the same data? Why show it twice? I would combine these figures as if Fig. 7 was a zoom-in of the sediment part of Fig. 6b and Fig. 6c. Please double check that the numbers match, I have the impression that, for example, Jan_2024 >51um POC flux shown in Fig. 6b is higher than when plotted in Fig. 7a

Response: Figure 6 shows the flux at 20 m, 40 m, 60 m, and the sediment accumulation flux.

Figure 7 shows the 60 m flux again to further emphasize (1) the quantity that is being available for deposition on the seafloor and (2) the difference between seasons.

> **L462-463**: It is confusing to say that the EQ depth was always ~4.5cm and then write that the majority of $Th_{xs,0}$ was confined to the top ~2-4.5cm. Also, looking at Figure 8 (please cite figure in that first sentence), the largest excess is found in the upper 1.5 cm.

Response: We thank the reviewer for pointing out this confusing phrasing. We have revised the text to cite Fig. 8 and to clarify that the majority of excess $^{234}$Th was concentrated in the upper ~2 cm (with the highest activities in the top 1.5 cm), and that equilibrium with supported $^{234}$Th was consistently reached by ~4.5 - 6 cm. The revised version of the text now reads – "Excess $^{234}$Th was concentrated in the surface sediments (upper ~2 cm; Fig. 8), with the highest activities observed in the top 1.5 cm. Excess $^{234}$Th decreased rapidly with depth, and equilibrium with supported $^{234}$Th was consistently reached by ~4.5 - 6 cm.

> - **L463**: "*Excess $^{234}$Th in the EQ was variable*". There should not be excess $^{234}$Th at the equilibrium depth. Do you mean above the EQ?

Response: Yes, we meant "above" the EQ and thank the reviewer for catching the mistake. We changed the manuscript accordingly.

**L467**: "*were consistent with core depth between 0 – 4 cm*". Not sure what do you mean. Are you referring to the concentrations and percentages found in those depths in other similar cores? Or do you mean that concentrations did not vary in the upper 4 cm of this study's cores? What concentrations and percentages of POC did you find between 4 cm and 6 cm? Was POC only measured in the upper 4 cm? There are POC:Th ratios deeper than 4 cm, were they extrapolated?

Response: Thank you for catching this. What we mean here is that the % POC concentrations were both consistent downcore (0 – 6 cm) and between cores. The text is now revised to include this. This consistency downcore and between cores was also found in Black et al. (2023) who observed that average % POC was statistically the same for all their core retrievals in different seasons. This consistency is also evident in historical coring efforts in the Bedford Basin (Buckley et al., 1995). The POC:$^{234}$Th ratios presented were not extrapolated.

> - **Figure 8**: Please polish the caption for panel b). The a) should be placed before "decay corrected". b) says "$POC:^{234}Th_{xs,0}$" but the title says "$POC:^{234}Th$". Also, no need to explain the supported in the caption, it has been explained in the main text and it is not shown in the figure.

Response: Change made as suggested.

> - **L559-560**: Could you expand on Lampitt (1985) statement? I understand the previous sentence about the fact that the ratios would be higher if there was resuspension of the top sediments, but I feel like Lampitt's sentence is not properly integrated in the discussion or lacking some more information. Not only the composition of the particles in the sediment-water interface facilitates resuspension, but hydrodynamics are also crucial, meaning that, even if those particles are

lithogenic and not detrital, with enough hydrodynamics they can be resuspended. Lampitt's 1985 study area is also 4000 m instead of the 70 m at Compass Station. Not sure it is the best comparison to make, unless you want to explain a bit more.

Response: We thank reviewer 1 for their insight here. We have since edit the manuscript to retain Lampitt's (1985) observation as a historical reference on particle composition facilitating resuspension but clarify that a direct comparison is not made. In section 4.2, we now describe more that our discussion of sediment resuspension is informed by both particulate β activities and local hydrodynamics. Although residual particulate β activities provides evidence of limited resuspension of the fine fraction, we now also include information about hydrodynamics & bottom currents (as you suggest). Recent physical oceanography measurements (Mali et al., 2024) report very weak near-bottom currents (typically < 1–2 cm s⁻¹) even during tidal cycles at the Compass Station. By contrast, studies in other shallow or coastal environments indicate that near-bed velocities of 20–30 cm s⁻¹ are generally required to erode bottom sediments (Ziervogel et al., 2021).

- **L567-569**: Boetius et al. 2013 talks about large algal chunks found at depth fresh and seen by cameras, not sure they refer to measuring fluorescence along the water column and finding high values at depth, so I am not sure the comparison applies. In any case, I would assume the deeper fluorescence signals found in this study are due to mixing, having some surface phytoplankton reaching deeper layers. Yet, blooms take place when water column presents higher stratification. Did you do isotopic analyses on the POC samples?

Response: We thank the reviewer for their comment. Boetius et al. (2013) report observations of large, fresh algal aggregates at depth rather than fluorescence profiles, and we have clarified the text to reflect this. The deeper fluorescence signals in our study are likely due to vertical mixing of surface phytoplankton. We did not perform isotopic analyses on the POC samples, and our discussion relies on bulk POC composition and depth profiles to infer particle transport and flux dynamics.

- **L567-573**: A high particle load is mentioned. Is that based on visual assessment (the water looked turbid) or there was some kind of measurement done that would indicate that (e.g., light penetration).

Response: We observed elevated chlorophyll fluorescence in the water column together with high particulate ²³⁴Th activities. To avoid ambiguity, we have since revised the text and specifically mention these measured parameters rather than using the general phrasing of "high particle load".

- **L580-582**: Bolanos et al. (2020) citation refers to importance of small phytoplankton in terms of biomass. Mention it.

Response: We have since revised text in manuscript (L580–582): "...small phytoplankton can play an important role in carbon export and biogeochemical cycling and have been shown to contribute substantially to biomass in the North Atlantic (Bolanos et al., 2020)."

- **L583-584**: Cite the reference for the *Synechococcus* statements.

Response: Change made. The citation for this statement is Bolanos et al., 2020.

- **Fig. 9**: Please clarify y-axis, I don't think I understand it and it is not described in the caption.

Response: We thank the reviewer for pointing this out. The y-axis represents normalized sample depth in the water column, calculated as the depth of the sample divided by the total water depth at the site. We use normalized depth instead of actual depth to emphasize vertical distance from the seafloor, a relevant metric in the context of sediment resuspension. For example, a sample taken at 20 m in a basin 70 m deep is plotted at 0.28 on the y-axis (i.e., 20/70). To clarify this, we have updated the y-axis label to "Normalized depth (sample depth / total water depth)" and revised the figure caption as follows:

"Profiles of $RA_{P234}$ are plotted versus normalized water depth (i.e., sample depth divided by total water depth, whereby 0 = surface, 1 = seafloor)."

**L605**: Just to clarify, the 8 dots that are above the detection limit had 0 $RA_{P234}$ from the large fraction, so those values are entirely from the small fraction particles or the difference from the detection limit value and the $RA_{P234}$ value in those cases is coming entirely from the small particles?

Response: The reported $RA_{P234}$ values represent the sum of both large and small particle fractions. In the eight instances where the $RA_{P234}$ values were above the detection limit, none of the large particle $RA_{P234}$ values were above the detection limit. Therefore, the $RA_{P234}$ signal originates entirely from the small particle fraction.

- **Section 4.4** I appreciate the authors revisiting Black et al (2023) POC:Th data and discussing the discrepancies observed. They could have avoided it and focus only on their current dataset, but instead they highlight the substantial differences and mention the potential sampling artifacts causing the extremely elevated ratios in that previous work.

Response: We agree with the reviewer that this discrepancy is important to the field. The large particle ratio is typically thought to be representative of sinking particles and is what used to report fluxes in many studies. We think that the data presented in the present study represent the first accurate measurements of large particle POC:$^{234}$Th ratios in the Bedford Basin.

- **L648**: Table 3 does not show other biogeochemical measurements; it shows estimates of POC export fluxes by others using other methods.

Response: We agree. In the revised manuscript, we have updated the table title to reflect its content more accurately and added a note to clarify the origin and method of the cited values.

- **L654:** Also similar to modelled POC flux reported in Black et al. 2023, based on the data shown in Table 3.

Response: Change made.

- **L662-663:** I obtain slightly different burial rate, which leads to slightly different burial efficiency. Minimal differences, but please double check the numbers.

Response: We thank the reviewer for double checking the numbers. Our reported burial efficiency of ~33% is derived from a mean depositional flux of $20 \pm 14$ mmol C m$^{-2}$ d$^{-1}$ and a burial flux of ~6.9 mmol C m$^{-2}$d$^{-1}$, which we calculated using the measured weight % POC (5.7%), mean sedimentation rate and dry bulk density. The reviewer's slightly different value likely arises from small differences in the chosen input parameters (e.g., rounding of sedimentation rate, bulk density, or POC%), or from comparing annualized fluxes rather than daily means.

- L671: Add at the end of the sentence "i.e., export efficiency". The following sentence starts talking about export efficiency and not ThE-ratio and might seem like they are different things for those not familiar with the thorium approach.

Response: We thank the reviewer for this helpful suggestion. We agree that clarifying these definitions will improve readability. In the revised manuscript, we only refer to this concept as "export efficiency" rather than using "ThE" and export efficiency.

- **L672-673**: It is the other way around, export/production = ThE; as stated in L676: "*divide the mean POC… by the mean NPP*"

Response: Agreed. Change made in revised manuscript.

- **L677**: 29.4 mmol C m$^{-2}$ d$^{-1}$ / 66 mmol C m$^{-2}$ d$^{-1}$ = 44.5% not 49% Please double check the numbers. L679 also mentions 49% ThE ratio and in the conclusions. Also check the burial efficiency reported in the conclusions (see my comment for L662-663).

Response: We thank the reviewer for catching this calculation error. The correct ratio is 29.4 mmol C m$^{-2}$ d$^{-1}$ / 66 mmol C m$^{-2}$ d$^{-1}$ = 0.445 (44.5%). We have since corrected this value in the results (L679), discussion and conclusions. The burial efficiency has likewise been double checked and is 33% (see above).

- **Table 3**:

    o The value reported by Hargrave and Taguchi (1978) is under "*Particulate flux*" and the units are in mmol C m$^{-2}$ d$^{-1}$. If it is not POC flux (otherwise I assume it would have said it) it should be total C to match the units reported. Is that the case? Please specify it in the title: "total C flux" or provide the necessary information as a table foot note.

Response: Hargrave & Taguchi (1978) report an annual mean sedimentation rate of ~ 17 mmol C m$^{-2}$ d$^{-1}$ which is based on total particulate carbon from sediment traps at 60 m in the Bedford Basin. In the revised manuscript, we have modified the table title to "Total Particulate Carbon Flux (mmol C m$^{-2}$ d$^{-1}$) and we included a footnote for the Hargrave & Taguchi (1978) number – "values from Hargrave and Taguchi (1978) represent total

particulate carbon sedimentation fluxes obtained from sediment trap measurements and are not limited to POC fluxes".

o The fluxes in the sediment reported in Black et al. 2023 are not shown in this table due to their extremely high values because of the particle sampling method used there, correct? It would be good to expand the first sentence of section 4.5 to explain the summary table 3, what was included in it and what not, and why.

Response: We thank the reviewer for this suggestion. Black et al. 2023 report sediment fluxes that averaged 2000 dpm m$^{-2}$ and ranged from 1100 – 3600 dpm m$^{-2}$. Indeed, the extremely high sediment fluxes reported in Black et al. (2023) are not included in our Table 3 because they result from the specific particle sampling method used in that study, which is not directly comparable to the approaches applied here. We have revised the first sentence of Section 4.5 to clarify what is included in Table 3 and the rationale for exclusions. The revised sentence now reads:

"Table 3 summarizes POC export fluxes derived in this study alongside comparable biogeochemical measurements in the Bedford Basin. Only measurements obtained using methodologies directly comparable to ours are included; the high sediment fluxes reported in Black et al. (2023), which likely arise from their distinctly different particle sampling approach, are excluded here to enable a meaningful comparison."

Technical comments:

- L124 Eq. 1 and L126 Eq 2.: Make sure the superscripts are shown as such for $^{238}U$ and $^{234}Th$

Response: Change made.

- **L166**: Eq. 5 uses Rap243 but later on, in L603-605 RAP243. Check, this term is used in other lines not cited here.

-

Response: The manuscript has been changed to consistently use $RA_{P234}$

- **L190**: Multiplied by

Response: Change made

- **L191**: Add (OC) after "Organic carbon" since it is used later in L194, or write it entirely in L194.

Response: Change made

- **Figure 2 panel E**: The image definition of this panel is worse than the others. Also, please provide the y-axes titles in the corresponding side of the graph. The legend is split but not the actual titles. The vertical line on the right side of the panel looks

odd. Can it be just the actual y-axis with its values? Also, the font size of the y-axes is much smaller than for the other panels. L281, T, S, O have not been defined before, add them in the previous line.

Response: We agree with the above comment. Panel E was made separate from A – D and merged to appear in the same format as A – D. The suggested changes are made for better clarity and conciseness to the format of A – D.

- **Figure 2**: Add depth [m] also in panels B, C and D.

Response: Change made

- **Table 1**: Nowhere in the text nor in the caption it is defined what LSF or SSF stands for.
  One can figure it out, but it would be good to add the whole wording in the caption.

Response: Edit made to now write out "large size fraction" and "small size fraction".

- **L404**: please cite the number of data point considered (p < 0.01, n= X)

Response: Change made.

- **Figure 5**: The legend is the same for both panels, even though the shape of the symbols is diZerent for large and small particles, correct? The same symbol could be used while highlighting the size fraction a bit more (maybe add it to the title?).

Response: Agreed. We use the same symbol in each panel and write "small size fraction" and "large size fraction" next to "POC:$^{234}$Th"

- **L425**: "*spring summer 2019*" missing an "and"

Response: Change made.

- **L436**: BB for Bedford Basin is only used here and in L657-658. I would just right the whole name in those lines.

Response: Change made.

- **L436**: "*...that this event brought additional…*" remove "it"

Response: Change made.

- **L474** cites only Fig.6b and 6c but it could also cite Fig. 7. Although, see my previous comment regarding the combination of these two figures in one.

Response: Change made.

- **Figure 8**. I don't mind the different symbols for the different sampling periods (although this is not done in the previous figures) but they seem to have different size? It could be an optical effect due to the different shape/color, but the dark blue

triangles seem bigger than the rest.

Response: Change made. We have since corrected for this so they appear consistent in size.

- **L511**: $^{234}Th$ appears as $234^{Th}$

Response: Change made.

- **L590**: It says "Jan 2022" but in all other instances the whole months it is written down, i.e., January. Same in L619 and L620.

Response: Edit made to fully write out "January"

- **L603-605**: Caption of Figure 9 – the residual activity here it is expressed as RAP234, different than in L166. Check the whole text and make sure to be consistent.

-

Response: Thank you for catching these details. Change made.

Work cited:

Black, E. E., Algar, C. K., Armstrong, M., & Kienast, S. S. (2023). Insights into constraining coastal carbon export from radioisotopes. *Frontiers in Marine Science*, *10*, 1254316. https://doi.org/10.3389/fmars.2023.1254316

Boetius, A., Albrecht, S., Bakker, K., Bienhold, C., Felden, J., Fernández-Méndez, M., Hendricks, S., Katlein, C., Lalande, C., Krumpen, T., Nicolaus, M., Peeken, I., Rabe, B., Rogacheva, A., Rybakova, E., Somavilla, R., Wenzhöfer, F., & RV Polarstern ARK27-3-Shipboard Science Party. (2013). Export of Algal Biomass from the Melting Arctic Sea Ice. *Science*, *339*(6126), 1430–1432. https://doi.org/10.1126/science.1231346

Bolaños, L. M., Karp-Boss, L., Choi, C. J., Worden, A. Z., Graff, J. R., Haëntjens, N., Chase, A. P., Della Penna, A., Gaube, P., Morison, F., Menden-Deuer, S., Westberry, T. K., O'Malley, R. T., Boss, E., Behrenfeld, M. J., & Giovannoni, S. J. (2020). Small phytoplankton dominate western North Atlantic biomass. *The ISME Journal*, *14*(7), 1663–1674. https://doi.org/10.1038/s41396-020-0636-0

B.T. Hargrave, G.A. Phillips, & S. Taguchi. (1976). *Sedimentation measurements in Bedford Basin, 1973-74*. Fish. Mar. Ser. Res. Dev. Rep. https://doi.org/10.13140/2.1.2815.3125

Buckley, D. E., Smith, J. N., & Winters, G. V. (1995). Accumulation of contaminant metals in marine sediments of Halifax Harbour, Nova Scotia: Environmental factors and historical trends. *Applied Geochemistry*, *10*(2), 175–195. https://doi.org/10.1016/0883-2927(94)00053-9

Lam, P. J., Ohnemus, D. C., & Auro, M. E. (2015). Size-fractionated major particle composition and concentrations from the US GEOTRACES North Atlantic Zonal Transect. *Deep Sea Research Part II: Topical Studies in Oceanography*, *116*, 303–320. https://doi.org/10.1016/j.dsr2.2014.11.020

Mali, S., Shan, S., Shore, J., & Crawford, A. (2024). Tide-Induced Bottom Current and Sediment Resuspension in Halifax Harbour. *Water*, *16*(22), 3272. https://doi.org/10.3390/w16223272

Ziervogel, K., Sweet, J., Juhl, A. R., & Passow, U. (2021). Sediment Resuspension and Associated Extracellular Enzyme Activities Measured ex situ: A Mechanism for Benthic-Pelagic Coupling in the Deep Gulf of Mexico. *Frontiers in Marine Science*, *8*, 668621. https://doi.org/10.3389/fmars.2021.668621

---

## Author Response (AR2)

**Edits made in accordance with referee questions/comments and editorial review**

September 30th, 2025

The authors would like to again thank both the 2 anonymous reviewers as well as the associate editor for their time and effort in reviewing this manuscript. When improving this manuscript, we have targeted suggestions that have been proposed by our reviewers, as well as the associate editor of this manuscript. In addition to the requested minor revisions, we have checked the whole text carefully and did some minor edits and clarifications to improve readability and flow. We have synthesized our point-by-point responses in black text and have provided the relevant changes to the manuscript in green italicized text below. Reviewer text is written in blue. Line numbers referenced in our responses correspond to the tracked changes version of the **updated manuscript**.

Note the remarks from the preceding review file validation: "1) Please ensure that the colour schemes used in your maps and charts allow readers with colour vision deficiencies to correctly interpret your findings. Please check your figures using the Coblis – Color Blindness Simulator (https://www.color-blindness.com/coblis-color-blindness-simulator/) and revise the colour schemes accordingly. --> Fig. 2 2) Your tables 1 to 3 contain coloured cells (grey shading). Please note that this will not be possible in the final revised version of the paper due to HTML conversion of the paper. For now, the process will continue, but please note that the final version cannot be published by using coloured tables."

The authors have since reviewed Fig 2 and have changed panel e) to a color-blind friendly pallet. Fig. 2 now complies to the majority of color vision deficiencies as confirmed through the simulator. We have also changed tables 1 to 3 to no longer contain colored cells.

Reviewer 1: (https://doi.org/10.5194/egusphere-2025-2493-RC1)

Reviewer 2: (https://doi.org/10.5194/egusphere-2025-2493-RC2)

**Reviewer 1 comments & author reply:**

This article provides quasi-seasonal coupled water column and surficial sediment sampling of Th-234 and POC to evaluate and refine the carbon budget in the Bedford Basin. I find the research necessary because, as highlighted by the authors, not many studies using the commonly applied approach of Th-234 to assess carbon export have been conducted in coastal areas. Additionally, the combination of water column and sediment measurements is not a usual approach either, yet it provides complementary data allowing to better understand the fate of the particle fluxes in coastal areas. By combining both types of sampling, the authors conclude that there is no major particle loss in the Bedford Basin and that the magnitude of the export and burial fluxes is similar to those reported in previous studies in coastal areas, which are relevant sites for organic carbon sequestration.

The paper is well presented; the text and the figures are clear and provide the necessary information to support their findings. I have specific comments regarding some aspects that need clarification or small corrections, but overall, the study is robust and contributes to the growing body of literature discussing carbon budgets in coastal environments.

**The authors would like to thank Reviewer 1 for their detailed insight and commentary on our manuscript.**

L111: It mentions that the water samples were collected with 10L Niskin bottles and that, after Th-230 was added, 2L were processed for total Th-234. I believe the way it is written can lead to confusion, since it might look like 10L were collected, spiked and then 2L subsampled, which I believe was not the case.

**Response:** In the revised manuscript, we clarify that 10 L Niskin bottles were used for sample collection, but only a 2 L subsample was spiked and processed for total 234Th measurements. The revised text now reads: "(L124) Water column samples were collected using 10 L Niskin bottles at 9 discrete depths. From each bottle, a 2 L subsample of unfiltered seawater was taken, spiked with a 230Th yield tracer, and processed according to the

L143: Please specify the pore size used to obtain filtered and UV treated seawater (no need to specify in that line the pore size of QMAs since it is mentioned in L141-142)

**Response:** The pore size used for seawater filtering was  $0.2 \, \mu m$ . This is now mentioned in L143 of the revised manuscript. We have also removed the repetition regarding the QMA pore size, which the reviewer correctly identified. The revised text now reads: (L166) "The large particle material (> 51  $\mu m$ ) captured on the Nitex screens was carefully rinsed onto QMA filters (1  $\mu m$  pore size, 25 mm diameter, pre-combusted) using 0.2  $\mu m$  filtered and ultraviolet (UV) treated seawater. The small particulate material (1 - 51  $\mu m$ ) captured on the QMA filter was subsampled by cutting out a 25 mm diameter punch.

L152-154: "For comparison, the POC:234Th ratio on the small particle fraction was also calculated using Eq. 4" is not correct. Please rephrase since Eq. 4 calculates POC flux, no ratios.

**Response:** We agree with reviewer 1 and have rephrased the relevant sentences. The POC flux can be calculated with either (a) the POC:234Th ratio on small particles or (b) the POC:234Th ratio on large particles using Eq. 4. We clarify in the manuscript that the fluxes reported using either a) or b) represent two alternative approaches to estimating flux, and that these should not be considered additive, but rather complimentary perspectives on export. The revised text now reads: "(L175) The POC flux can be calculated with either a) the POC:234Th ratio on small particles, or b) the POC:234Th ratio on large particles using Eq. 4. The fluxes reported using either a) or b) represent two alternative approaches to estimating flux, and should not be considered additive, but rather complimentary perspectives on export."

L151-154 and Eq. 4 can be simplified by adding a subscript next to the ration in the formula so that the text just needs to say that POC fluxes are calculated using Eq. 4 using the ratios of the different particle fractions.

**Response:** We have since tightened this sentence to be clearer but have added subscripts next to Eq. 4 for clarity.

L157-165: The first two sentences seem a bit repetitive and can be streamlined. I think the repetition of "novel proxy and residual beta activity" makes it look repetitive. Use the "abbreviation" Rap243 instead after defining it at the beginning.

**Response:** Change made as suggested. The revised text now reads: (L182) "To investigate the role of resuspension in this study and in the Bedford Basin as a whole, the residual beta activity ( $Ra_{P234}$ ) was analyzed.  $Ra_{P234}$  is a novel resuspension proxy (Lin et al., 2016) and has previously been applied to investigate particle dynamics in coastal sites."

L169: Can you provide the uncertainty of np2 and the blank? These blank counts come from the dipped filters, correct? Also in that line, notice that cpm was used before and defined later.

**Response:** We thank reviewer 1 for pointing out these issues. As the reviewer correctly suggests, the blank counts (n0) were obtained from dipped filters. Dipped filters are deployed with the pump to depth, but, importantly, no seawater is pumped through them. They can be considered a seawater process blank and are often referred to as a "dipped blank", following Lam et al., 2015. We now report the uncertainties for both nP2 and n0 based on counting statistics. The revised sentence reads: (L193) "Where  $n_{p2}$  is the second  $\beta$  counting rate of particulate  $^{234}Th$  (mean  $0.3 \pm 0.07$  counts per minute (cpm)),  $n_0$  is the blank mean ( $0.24 \pm 0.02$  cpm),  $\varepsilon$  is the detector efficiency (0.38), and V is volume of seawater pumped (E). The detection limit is defined here as two times the standard deviation of the blank and corresponds to 128.2 dpm  $e^{-3}$ .

**Section 2.5** Excess 234Th inventories requires more detail:**

**Response:** We have now revised section 2.5 in the manuscript to include more details on the method & processing of the sediments. The details are explained below.

How thick were the sections of the cores (0.5 cm, 1 cm) and how far down where they cut/measured (6 cm, based on

**Response:** Cores were generally sliced into 0.5 cm thick sections. In some upper core intervals, we took 1 cm thick sections, as the sediment was too soft to cut thinner sections. To clarify, all cores were measured for 234Th down to 10 cm, some cores were only measured down to 6 cm for TOC. Ideally, we wanted to slice cores into 0.5 cm samples for higher resolution in the upper most sediments. If this was not possible, the cores were sectioned into 1 cm samples.

How quickly were the cores cut after collection? Were they kept in a freezer until processing?

**Response:** Cores were either sliced on the same day they were retrieved in the field or stored in a walk-in fridge (4°C) for 1-2 days (max) prior to slicing. The revised text now reads: (L199) "Sediments were processed either the same day of retrieval or stored at (4°C) for 24-48 hrs prior to processing"

How were they dried? For how long approx.? Was the weight checked multiple times to ensure that the sediment was dry?

**Response:** The sediments were dried at 55°C in our laboratory oven for approximately 24 hours. After this period, they were weighed and returned to oven and reweighed at regular intervals until a constant weight was achieved. The revised text now reads: (L200) "Cores were sliced in 0.5 - 1 cm thick sections and then dried at 55°C for approximately 24 hrs. After this period, they were weighed and returned to the oven and reweighed at regular intervals until a constant weight was achieved."

Was the dry weight corrected for salt content?

**Response:** Yes, the dry weights were corrected for salt content. Salt contribution from porewater was estimated based on salinity measurements of the overlying water and subtracted from the measured dry weight. This clarification has been added to the method section of the revised manuscript. The revised text now reads: (L202) "To correct for residual salt content, salt contribution from porewater was estimated from the salinity of the overlying water and subtracted from the measured dry weight."

How much sediment was approximately used for the measurement?

**Response:** Approximately 15 g of dry sediment was used for gamma counting. The revised text now reads: (L203) "Once dry, sediments were manually homogenized and weighed into a petridish. Analysis of sedimentary 234Th and 238U activities were carried out by non-destructive gamma counting on High-Purity Germanium (HPGe) detectors (Dartmouth College, USA) with ~ 15 g of sediment."

How long were the samples measured for and how long after collection (since Th- 234 has a 24.1 d half-life)?

**Response:** Gamma analysis was typically within less than 24 days, i.e., one half-life of sediment retrieval in the field. Samples were counted for 7 to 24 h depending on the time to achieve counting uncertainties of less than 5 % on the primary peaks.

How was the mean supported activity obtained? Did you remeasure all the sections or just certain ones? L465 suggests only section 4.5 to 6 was used as supported. Please provide more details.

**Response:** We thank reviewer 1 for this comment. All cores were measured for  $^{234}$ Th down to 10 cm. The mean supported  $^{234}$ Th activity was determined from the deepest sediment intervals in each core where no excess  $^{234}$ Th was detectable, i.e., below the penetration depth of excess  $^{234}$ Th. For our cores, this was consistently at  $\sim$ 4.5–6 cm depth (i.e. at 4.5–6 cm, the  $^{234}$ Th activities were consistent and showed no further decrease in  $^{234}$ Th activity). These deepest sections were measured directly and used to define the supported activity, which was

then subtracted from the total 234Th activities in the overlying sediments to calculate excess 234Th. We have now clarified this in the methods section. (L207) "Supported 234Th, in equilibrium with 238U present in the sediment, is identified at the core depth below which no further decrease in 234Th activities is observed ("Equilibrium Depth"; EQ). The supported activity is subtracted from the total activity to derive  $^{234}Th_{xs,0}$  (Eq. 6)."

What type of gamma detector was used? The gamma counting was done using the petri-dish, so it was not done in a well-type gamma counter, right? Mention it.

**Response:** Correct. Gamma counting was conducted using Canberra Intrinsic High-Purity Germanium (HPGe) detectors in a planar configuration, not a well-type of detector. Samples were counted in low-background geometry using petri dishes, which were placed directly on the detector surface to optimize efficiency for low-energy gamma emissions (such as the 63.3 keV peak of 234Th). We have added this information in the revised paper. (L203) "Analysis of sedimentary 234Th and 238U activities were carried out by non-destructive gamma counting on High-Purity Germanium (HPGe) detectors (Dartmouth College, USA) with ~ 15 g of sediment".

L191-192: Please explain how the organic carbon samples were processed before measurements. Were they also fumed with HCl, as per filters?

**Response:** We did not fume the sediment samples with HCl prior to CN analysis, and spell this out in the revised text. (L220) "Previous work on Bedford Basin sediments compared acid-fumigated and non-acid fumigated samples and found that the mean carbon content did not differ between the acid fumigated (5.5%  $\pm$  0.5%, n=29) and non-acid fumigated (5.7%  $\pm$  0.8%, n=39) samples (Black et al., 2023). This published finding is consistent with additional unpublished measurements that were made over the years at Dalhousie University. Collectively, the data imply that most of the sedimentary carbon is organic in nature, and thus acid fumigation was not done here."

**L205:** Please provide a threshold of Chl-a or some definition for the blooms in the region or a usual range of Chl-a values during blooms. The text reads that a 11.1 mg/m3 chl-a signal is a weak bloom but would be good to have some sort of a scale to assess that.

**Response:** Based on observations made by the Bedford Basin Monitoring Program since 1993, "background" chlorophyll-a typically ranges from ~3 to 10 mg m-3 (surface, 1–10 m) in the summer. In contrast, spring blooms range from ~7 to 10 mg m-3 and up to ~28 mg m-3. Extreme short-lived events have reached chlorophyll concentrations over 100 mg m-3. Therefore, we define a "weak bloom" in Bedford Basin as chlorophyll-a concentrations just above the summer background (e.g., >10 mg m-3) but below intense bloom threshold of 28 mg m-3. We will include these ranges and citations to make the terminology clearer in a revised manuscript. The revised text now reads: "(L63) Chlorophyll-a (chl-a) concentrations in surface waters (1 – 10 m) typically range from ~3 – 10 mg m-3. Spring blooms typically range from ~7 – 28 mg m-3, (Fisheries and Oceans Canada, 2025), while extreme short-lived events can exceed 100 mg m-3 (Li & Dickie., 2001). "

**Figure 4**: "Total particulate 234Th is equal to the sum of the large and small particles" Why is that sentence there for Th-234 and not for POC? Shouldn't the y- axes just be Particulate POC and Particulate 234Th, since there are different size fractions represented in both plots?

Response: We agree that axis labels and definitions should reflect the size-fractionation. In our study, particulate material was collected sequentially on a 51  $\mu m$  screen and a 1  $\mu m$  filter from the same pump casts. After corrections for blanks, yield (for  $^{234}$ Th), and volume,

we define total particulate  $^{234}$ Th (>1  $\mu$ m) as the sum of  $^{234}$ Th (1–51  $\mu$ m) and  $^{234}$ Th (>51  $\mu$ m). Similarly, total particulate POC (>1  $\mu$ m) is defined as the sum of POC (1–51  $\mu$ m) and POC (> 51  $\mu$ m). We have updated the figure axes to read "Particulate  $^{234}$ Th (>1  $\mu$ m)" and "Particulate POC (> 1  $\mu$ m)". The legend now also correctly identifies the 1–51  $\mu$ m and >51  $\mu$ m fractions. Revised figure caption now reads: (*L291*) "*Figure 4. Size-fractionated* (1–51  $\mu$ m, > 51  $\mu$ m) pools of (a) particulate organic carbon (POC,  $\mu$ M) and (b) total particulate  $^{234}$ Th (dpm  $^{-1}$ ) at ~20, 40, and 60 m for each sampling event. Bars are stacked; totals equal the sum of the small and large fractions (legend shows % contribution by size class). Numbers above bars indicate sampling depth; dashed lines separate events. The small fraction generally dominates both POC and  $^{234}$ Th"

**L431**: "POC export fluxes (Fig. 6) were calculated following Eq. 4 and using >51um ratios (Fig. 5a)" it's correct, but not complete, because the export fluxes shown in Fig. 6 are also calculated using the 1-51um ratios from Fig. 5b. Please refine this paragraph to streamline it. You could cite both figures and then start talking about the small fraction fluxes, were no pattern with depth was observed, and then talk about the large fraction fluxes, since they are discussed a bit more.

**Response:** We thank the reviewer for this suggestion. We have revised the paragraph to clarify that fluxes were calculated using both the small (1–51  $\mu$ m) and large (>51  $\mu$ m) POC:234Th ratios (Figs. 5b and 5a, respectively). We now introduce both fractions, then describe the small fraction first, followed by the large fraction, where more variability was observed. L431 now reads: (L323) "POC export fluxes (Fig. 6) were calculated following Eq. 4 using both the 1–51  $\mu$ m and >51  $\mu$ m POC:23Th ratios (Figs. 5b and 5a, respectively). Fluxes based on the small size fraction (1–51  $\mu$ m) showed no consistent pattern with depth, while fluxes from the large size fraction (>51  $\mu$ m) varied more strongly. At the deepest depth sampled (60–65 m), large fraction POC fluxes had a mean (± s.d.) of 20 ± 14 mmol C m-2 d-1 (Fig. 7a) and ranged from 3.6 ± 1.1 to 44.5 ± 4.1 mmol C m-2 d-1."

**Figure 6**: The a) should be moved in front of "*Profiles of cumulative*...." since "*Bedford Basin 1D steady state fluxes in 2021-2024*" applies to the three panels. Make the y-axes to match between the three panels, otherwise it creates a visual distortion as if the POC ones were deeper. Why all the different symbols in the Th-234 flux? Pick one, as done for the POC fluxes based on particle size. Also, the R2 shown in graphic a) is not explained. Maybe add the whole linear regression and cite the figure in the main text (L428).

**Response:** We thank the reviewer for their insightful comments and keen eye. We have since made the appropriate changes by making the y-axis consistent and choosing one symbol. The R squared value comes from the linear regression of all dates. This R-squared value will be added into the text of the manuscript. "(L316) Cumulative 234Th fluxes increase nearly linearly with water depth ( $R^2 = 0.95$ ) and do not show much seasonal variation (Fig. 6a)."

I am a bit confused regarding the information provided in Fig. 6 in the sediment part of panels b) and c) and what it is shown in Fig. 7. Isn't it the same data? Why show it twice? I would combine these figures as if Fig. 7 was a zoomin of the sediment part of Fig. 6b and Fig. 6c. Please double check that the numbers match, I have the impression that, for example, Jan 2024 >51um POC flux shown in Fig. 6b is higher than when plotted in Fig. 7a

**Response:** Figure 6 shows the flux at 20 m, 40 m, 60 m, and the sediment accumulation flux. Figure 7 shows the 60 m flux again to further emphasize (1) the quantity that is being available for deposition on the seafloor and (2) the difference between seasons.

L462-463: It is confusing to say that the EQ depth was always  $\sim$  4.5cm and then write that the majority of Thxs,0 was confined to the top  $\sim$ 2-4.5cm. Also, looking at Figure 8 (please cite figure in that first sentence), the largest excess is found in the upper 1.5 cm.

**Response:** We thank the reviewer for pointing out this confusing phrasing. We have revised the text to cite Fig. 8 and to clarify that the majority of excess  $^{234}$ Th was concentrated in the upper  $\sim$ 2 cm (with the highest activities in the top 1.5 cm), and that equilibrium with supported  $^{234}$ Th was consistently reached by  $\sim$ 4.5 - 6 cm. The revised version of the text now reads -(L353) "Excess  $^{234}$ Th was concentrated in the surface sediments, with the highest activities observed in the top 1.5 cm (Fig. 8). Excess  $^{234}$ Th decreased with depth, and equilibrium with supported  $^{234}$ Th was consistently reached by 4.5-6 cm in every core."

**L463**: "Excess 234Th in the EQ was variable". There should not be excess 234Th at the equilibrium depth. Do you mean above the EQ?

**Response:** Yes, we meant "above" the EQ and thank the reviewer for catching the mistake. We changed the manuscript accordingly. The revised text now reads (L354) "Excess 234Th above the EQ was variable in each sampling event, averaging  $15.3 \pm 16$  dpm g-1 and ranging from 4.8-46 dpm  $g^{-1}$  (Fig. 8a)."

L467: "were consistent with core depth between 0-4 cm". Not sure what do you mean. Are you referring to the concentrations and percentages found in those depths in other similar cores? Or do you mean that concentrations did not vary in the upper 4 cm of this study's cores? What concentrations and percentages of POC did you find between 4 cm and 6 cm? Was POC only measured in the upper 4 cm? There are POC:Th ratios deeper than 4 cm, were they extrapolated?

**Response:** Thank you for catching this. What we mean here is that the % POC concentrations were both consistent downcore (0-6 cm) and between cores. The text is now revised to include this. This consistency downcore and between cores was also found in Black et al. (2023) who observed that average % POC was statistically the same for all their core retrievals in different seasons. This consistency is also evident in historical coring efforts in the Bedford Basin (Buckley et al., 1995). The revised text now reads: (L357) "POC concentrations were consistent both downcore (0-6 cm) and between cores (n=5), in line with previous studies in the Bedford Basin (Black et al., 2023; Buckley et al., 1995)."

**Figure 8**: Please polish the caption for panel b). The a) should be placed before "decay corrected". b) says "*POC*.234*Thxs*,0" but the title says "*POC*.234*Th*". Also, no need to explain the supported in the caption, it has been explained in the main text and it is not shown in the figure.

**Response:** Change made as suggested. The revised text now reads: (L380) "Sediment downcore profile of (a) decay corrected excess 234Th (234Thxs,0), derived from subtracting the supported 234Th from decay corrected 234Th in sediments."

L559-560: Could you expand on Lampitt (1985) statement? I understand the previous sentence about the fact that the ratios would be higher if there was resuspension of the top sediments, but I feel like Lampitt's sentence is not properly integrated in the discussion or lacking some more information. Not only the composition of the particles in the sediment-water interface facilitates resuspension, but hydrodynamics are also crucial, meaning that, even if those particles are lithogenic and not detrital, with enough hydrodynamics they can be resuspended. Lampitt's 1985 study area is also 4000 m instead of the 70 m at Compass Station. Not sure it is the best comparison to make, unless you want to explain a bit more.

**Response:** We thank reviewer 1 for their insight here, we have since removed the Lampitt statement as we agree it is not the best comparison to make. In section 4.2, we now describe more that our discussion of sediment resuspension is informed by both particulate  $\beta$  activities and local hydrodynamics. Although residual particulate  $\beta$  activities provides evidence of limited resuspension of the fine fraction, we now also include information about hydrodynamics & bottom currents (as you suggest). Recent physical oceanography measurements (Mali et al., 2024) report very weak near-bottom currents (typically < 1–2 cm s-1) even during tidal cycles at the Compass Station. By contrast, studies in other shallow or coastal environments indicate that near-bed velocities of 20–30 cm s-1 are generally required to erode bottom sediments (Ziervogel et al., 2021).

The revised text now reads: (L415) "Resuspension from the seafloor could be a source of particles to the water column, though multiple lines of evidence indicate it is likely minimal at the Compass Station. First, sedimentary POC:  $^{234}Th$  ratio is an order of magnitude higher than those in the water column (Fig 8b), likely due to the temporal decay of  $^{234}Th$  exceeding the remineralization of POC. Bottom resuspension would be expected to introduce particles with these elevated POC:  $^{234}Th$  which is not reflected in deep water column POC:  $^{234}Th$  ratios from this study. Second, recent measurements of physical properties in the Bedford Basin report very weak nearbottom currents (typically < 1 - 2 cm  $s^{-1}$ ), even during tidal cycles at the Compass Station. Other coastal studies in shallow systems have indicated that near-bed velocities of 20 - 30 cm  $s^{-1}$  are generally required to erode bottom sediments (Ziervogel et al., 2021). These findings support negligible resuspension from the seafloor at the Compass Station."

**L567-569**: Boetius et al. 2013 talks about large algal chunks found at depth fresh and seen by cameras, not sure they refer to measuring fluorescence along the water column and finding high values at depth, so I am not sure the comparison applies. In any case, I would assume the deeper fluorescence signals found in this study are due to mixing, having some surface phytoplankton reaching deeper layers. Yet, blooms take place when water column presents higher stratification. Did you do isotopic analyses on the POC samples?

**Response:** We thank the reviewer for their comment. Boetius et al. (2013) report observations of large, fresh algal aggregates at depth rather than fluorescence profiles, and we have clarified the text to reflect this. The deeper fluorescence signals in our study are likely due to vertical mixing of surface phytoplankton. We did not perform isotopic analyses on the POC samples, and our discussion relies on bulk POC composition and depth profiles to infer particle transport and flux dynamics. The revised text now reads: "(L427) Furthermore, deep and extensive CTD-derived chl-a fluorescence signals (Fig. 2d) suggest that biological processes could also be responsible for particle creation and continued scavenging in deeper waters. These features are not unique (Black et al., 2023), and fluorescent organic material has been visually observed at depths > 3000 m in the Arctic as a result of a rapid bloom-collapse cycle (Boetius et al., 2013)."

**L567-573**: A high particle load is mentioned. Is that based on visual assessment (the water looked turbid) or there was some kind of measurement done that would indicate that (e.g., light penetration).

**Response:** We observed elevated chlorophyll fluorescence in the water column together with high particulate 234Th activities. To avoid ambiguity, we have since revised the text and specifically mention these measured parameters rather than using the general phrasing of "high particle load". The revised text now reads: (L430) "While additional data (e.g. Underwater Vision Profiler, net primary production (NPP), backscatter, etc.) are needed to identify the source for the evident fluorescence signals at depths much deeper than expected given the high particle load (i.e. elevated chl-a and POC: 234Th)"

L580-582: Bolanos et al. (2020) citation refers to importance of small phytoplankton in terms of biomass. Mention it.

**Response:** We have since revised text in manuscript: "(L441) Small phytoplankton ( $< 20 \,\mu m$ ) can play an important role in carbon export and biogeochemical cycling and have been shown to contribute substantially to biomass in the North Atlantic, with some species especially dominant in subpolar regions during winter months (Bolaños et al., 2020)"

**L583-584**: Cite the reference for the *Synechococcus* statements.

**Response:** Change made. The citation for this statement is Bolanos et al., 2020.

Fig. 9: Please clarify y-axis, I don't think I understand it and it is not described in the caption.

**Response:** Thank you for pointing this out. The y-axis represents normalized sample depth in the water column, calculated as the depth of the sample divided by the total water depth at that site. We do this instead of just depth to emphasize distance from seafloor, since we are speaking of resuspension from sediments. For example, a sample taken at 20 m in a basin 70 m deep would be plotted at 0.28 on the y-axis. To clarify this, we have updated the y-axis label to "Normalized depth (sample depth / total water depth)" and revised the figure caption as follows: (L456) "Profiles of RAP234 versus normalized depth in the water column, calculated as sample depth divided by total water depth (0 = surface, 1 = bottom)."

**L605**: Just to clarify, the 8 dots that are above the detection limit had 0 RAP234 from the large fraction, so those values are entirely from the small fraction particles or the difference from the detection limit value and the RAP234 value in those cases is coming entirely from the small particles?

**Response:** The reported RAP234 values represent the sum of both large and small particle fractions. In the eight instances where the RAP234 values were above the detection limit, none of the large particle RAP234

values were above the detection limit. Therefore, the RAP234 signal originates entirely from the small particle fraction.

**Section 4.4** I appreciate the authors revisiting Black et al (2023) POC:Th data and discussing the discrepancies observed. They could have avoided it and focus only on their current dataset, but instead they highlight the substantial differences and mention the potential sampling artifacts causing the extremely elevated ratios in that previous work.

**Response:** We agree with the reviewer that this discrepancy is important to the field. The large particle ratio is typically thought to be representative of sinking particles and is what used to report fluxes in many studies. We think that the data presented in the present study represent the first accurate measurements of large particle POC: 234Th ratios in the Bedford Basin.

L648: Table 3 does not show other biogeochemical measurements; it shows estimates of POC export fluxes by others using other methods.

**Response:** We agree. In the revised manuscript, we have updated the table title to reflect its content more accurately and added a note to clarify the origin and method of the cited values. The revised text now reads: (L557) "An updated carbon budget of the Bedford Basin. Carbon fluxes derived in this study alongside comparable carbon flux measurements in the Bedford Basin".

L654: Also similar to modelled POC flux reported in Black et al. 2023, based on the data shown in Table 3.

Response: change made.

L662-663: I obtain slightly different burial rate, which leads to slightly different burial efficiency. Minimal differences, but please double check the numbers.

**Response:** We thank the reviewer for double checking the numbers. Our reported burial efficiency of  $\sim 33\%$  is derived from a mean depositional flux of  $20 \pm 14$  mmol C m-2 d-1 and a burial flux of  $\sim 6.9$  mmol C m-2d-1, which we calculated using the measured weight % POC (5.7%), mean sedimentation rate and dry bulk density. The reviewer's slightly different value likely arises from small differences in the chosen input parameters (e.g., rounding of sedimentation rate, bulk density, or POC%), or from comparing annualized fluxes rather than daily means.

L671: Add at the end of the sentence "i.e., export efficiency". The following sentence starts talking about export efficiency and not ThE-ratio and might seem like they are different things for those not familiar with the thorium approach.

**Response:** We thank the reviewer for this helpful suggestion. We agree that clarifying these definitions will improve readability. In the revised manuscript, we only refer to this concept as "export efficiency" rather than using "ThE" and export efficiency. The revised text now reads: (L545) "The thorium export efficiency ratio (ThE-ratio) is the ratio of POC flux (based on 234Th measurements) relative to NPP in the euphotic zone. To estimate the ThE ratio in the Bedford Basin, we divide the annual mean primary production in the basin by the annual mean flux of POC measured in this study".

**L672-673**: It is the other way around, export/production = ThE; as stated in L676: "divide the mean POC... by the mean NPP"

**Response:** Agreed. Change made in revised manuscript. The revised text now reads: (L545) "The thorium export efficiency ratio (ThE-ratio) is the ratio of POC flux (based on 234Th measurements) relative to NPP in the euphotic zone. To estimate the ThE ratio in the Bedford Basin, we divide the annual mean flux of POC measured in this study by the annual mean primary production in the basin. Primary production is rarely measured directly in the Bedford Basin with only one record of NPP from incubation studies (50 mmol C m-2 d-1; Platt, 1975). Black et al (2023) approximated an annual mean NPP of 66 mmol C m-2 d-1 using a model based on in-situ chl-a and PAR data. This NPP estimate is within the range of NPP estimates from the nearby Gulf of Maine (i.e., 26-399 mmol C m-2 d-1; Charette et al., 2001). With this information, mean POC export at the base of the euphotic zone (~20 m; 29.4 mmol C m-2 d-1) is divided by mean NPP to and derives ThE ratios of 44.5%."

L677: 29.4 mmol C  $m^{-2} d^{-1} / 66$  mmol C  $m^{-2} d^{-1} = 44.5\%$  not 49% Please double check the numbers. L679 also mentions 49% ThE ratio and in the conclusions. Also check the burial efficiency reported in the conclusions (see my comment for L662-663)

**Response:** We thank the reviewer for catching this calculation error. The correct ratio is 29.4 mmol C  $m^{-2}$   $d^{-1}$  / 66 mmol C  $m^{-2}$   $d^{-1}$  = 0.445 (44.5%). We have since corrected this value in the results (L552), discussion and conclusions. The burial efficiency has likewise been double checked and is 33% (see above).

**Table 3**: The value reported by Hargrave and Taguchi (1978) is under "*Particulate flux*" and the units are in mmol C m-2 d-1. If it is not POC flux (otherwise I assume it would have said it) it should be total C to match the units reported. Is that the case? Please specify it in the title: "total C flux" or provide the necessary information as a table foot note.

**Response:** Hargrave & Taguchi (1978) report an annual mean sedimentation rate of ~ 17 mmol C m-2 d-1 which is based on total particulate carbon from sediment traps at 60 m in the Bedford Basin. In the revised manuscript, we have modified the table to add subscripts beneath it. It now reads: (L562) "Total particulate carbon flux from cylindrical trap estimates (Hargrave and Taguchi, 1978), not limited to POC; b Model from Rakshit et al., 2025, c Model from Black et al., 2023."

The fluxes in the sediment reported in Black et al. 2023 are not shown in this table due to their extremely high values because of the particle sampling method used there, correct? It would be good to expand the first sentence of section 4.5 to explain the summary table 3, what was included in it and what not, and why.

**Response:** We thank the reviewer for this suggestion. Black et al. 2023 report sediment fluxes that averaged 2000 dpm m-2 and ranged from 1100 – 3600 dpm m-2. Indeed, the extremely high sediment fluxes reported in Black et al. (2023) are not included in our Table 3 because they result from the specific particle sampling method used in that study, which is not directly comparable to the approaches applied here. We have revised the table caption as: (L557) "Table 3. An updated carbon budget of the Bedford Basin. Carbon fluxes derived in this study alongside comparable carbon flux measurements in the Bedford Basin. The measured fluxes reported in Black et al. (2023), which likely arise from their different particle sampling approach, are excluded here to enable a meaningful comparison."

Technical comments:

L124 Eq. 1 and L126 Eq 2.: Make sure the superscripts are shown as such for 238U and 234Th

Response: Change made

L166: Eq. 5 uses Rap243 but later on, in L603-605 RAP243. Check, this term is used in other lines not cited here.

Response: Change made. The manuscript has been changed to consistently use RAP234

L190: Multiplied by:

Response: Change made

L191: Add (OC) after "Organic carbon" since it is used later in L194, or write it entirely in L194.

**Response:** Change made

Figure 2 panel E: The image definition of this panel is worse than the others. Also, please provide the y-axes titles in the corresponding side of the graph. The legend is split but not the actual titles. The vertical line on the right side of the panel looks odd. Can it be just the actual y-axis with its values? Also, the font size of the y-axes is much smaller than for the other panels. L281, T, S, O have not been defined before, add them in the previous line.

**Response:** We agree with the above comment. Panel E was made separate from A - D and merged to appear in the same format as A - D. The suggested changes are made for better clarity and conciseness to the format of A - D.

**Figure 2**: Add depth [m] also in panels B, C and D.

**Response:** Change made.

**Table 1**: Nowhere in the text nor in the caption it is defined what LSF or SSF stands for. One can figure it out, but it would be good to add the whole wording in the caption.

Response: Edit made to now write out "large size fraction" and "small size fraction".

**L404**: please cite the number of data point considered (p < 0.01, n = X)

**Response:** Change made. The revised text now reads "On average, higher POC:  $^{234}$ Th ratios on the larger particles were observed compared to the smaller particles (p < 0.01, n = 18)"

**Figure 5**: The legend is the same for both panels, even though the shape of the symbols is different for large and small particles, correct? The same symbol could be used while highlighting the size fraction a bit more (maybe add it to the title?).

**Response:** Agreed. We use the same symbol in each panel and write "small size fraction" and "large size fraction" next to "POC:234Th"

L425: "spring summer 2019" missing an "and"

Response: Change made.

**L436**: BB for Bedford Basin is only used here and in L657-658. I would just right the whole name in those lines.

**Response:** Change made.

Response: Change made.

L474 cites only Fig.6b and 6c but it could also cite Fig. 7. Although, see my previous comment regarding the combination of these two figures in one.

Response: Change made.

**Figure 8**. I don't mind the different symbols for the different sampling periods (although this is not done in the previous figures) but they seem to have different size? It could be an optical effect due to the different shape/color, but the dark blue triangles seem bigger than the rest.

**Response:** Change made. We have since corrected for this by making all the symbols the same to be consistent.

**L511**: 234Th appears as 234Th

Response: Change made.

**L590**: It says "Jan 2022" but in all other instances the whole months it is written down, i.e., January. Same in L619 and L620.

Response: Edit made to fully write out "January"

**L603-605**: Caption of Figure 9 – the residual activity here it is expressed as RAP234, different than in L166. Check the whole text and make sure to be consistent.

Response: Thank you for catching these details. Change made.

**Work cited:**

- Black, E. E., Algar, C. K., Armstrong, M., & Kienast, S. S. (2023). Insights into constraining coastal carbon export from radioisotopes. *Frontiers in Marine Science*, 10, 1254316. https://doi.org/10.3389/fmars.2023.1254316
- Boetius, A., Albrecht, S., Bakker, K., Bienhold, C., Felden, J., Fernández-Méndez, M., Hendricks, S., Katlein, C., Lalande, C., Krumpen, T., Nicolaus, M., Peeken, I., Rabe, B., Rogacheva, A., Rybakova, E., Somavilla, R., Wenzhöfer, F., & RV Polarstern ARK27-3-Shipboard Science Party. (2013). Export of Algal Biomass from the Melting Arctic Sea Ice. *Science*, 339(6126), 1430–1432. https://doi.org/10.1126/science.1231346
- Bolaños, L. M., Karp-Boss, L., Choi, C. J., Worden, A. Z., Graff, J. R., Haëntjens, N., Chase, A. P., Della Penna, A., Gaube, P., Morison, F., Menden-Deuer, S., Westberry, T. K., O'Malley, R. T., Boss, E., Behrenfeld, M. J., & Giovannoni, S. J. (2020). Small phytoplankton dominate western North Atlantic biomass. *The ISME Journal*, 14(7), 1663–1674. https://doi.org/10.1038/s41396-020-0636-0
- B.T. Hargrave, G.A. Phillips, & S. Taguchi. (1976). *Sedimentation measurements in Bedford Basin, 1973-74*. Fish. Mar. Ser. Res. Dev. Rep. <a href="https://doi.org/10.13140/2.1.2815.3125">https://doi.org/10.13140/2.1.2815.3125</a>
- Buckley, D. E., Smith, J. N., & Winters, G. V. (1995). Accumulation of contaminant metals in marine sediments of Halifax Harbour, Nova Scotia: Environmental factors and historical trends. *Applied Geochemistry*, *10*(2), 175–195. <a href="https://doi.org/10.1016/0883-2927(94)00053-9">https://doi.org/10.1016/0883-2927(94)00053-9</a>
- Lam, P. J., Ohnemus, D. C., & Auro, M. E. (2015). Size-fractionated major particle composition and concentrations from the US GEOTRACES North Atlantic Zonal Transect. *Deep Sea Research Part II: Topical Studies in Oceanography*, *116*, 303–320. https://doi.org/10.1016/j.dsr2.2014.11.020

**Reviewer 2 comments & author reply:**

The study by Healey et al., investigates the water column  $^{234}$ Th  $-\,^{238}$ U disequilibrium and surficial sediment inventory of excess  $^{234}$ Th to constrain the particulate organic carbon (POC) export and burial in Bedford Basin. The integration of water column and sediment data to quantify the POC fluxes is appreciable.

Authors conducted quasi-season sampling from 2021 - 2024, capturing the seasonal dynamics over three-year period. The water column and sedimentary 234Th fluxes and POC fluxes along with bottom sediment depositional POC fluxes are in close agreement which verifies the estimates of POC fluxes in water column and mostly contributed by the vertical setting of particles. The study is comprehensive with proper methodology.

The authors thank Reviewer 2 for their thoughtful insight into our manuscript, especially in further developing our Discussion regarding non steady state terms in the Bedford Basin.

Though the Bedford Basin is relatively small, the data from a single central station might not capture the lateral heterogeneity or edge effects. In the center, as shown by the data, the POC contribution is mostly from the primary production or vertical setting of particles, however, the lateral contribution of the particles cannot be ignored, especially in this shallow fjord. It can be discussed in the manuscript in general regarding the overall basin for wider implications.

Response: Indeed, our data are from a single, centrally located station in the Bedford Basin and we agree with the reviewer that lateral heterogeneity could potentially contribute to 234Th and POC dynamics in this shallow fjord system. However, we anticipate that lateral impacts are small, as explained below. In the revised manuscript, we have added a statement in the discussion (section 4) that (a) acknowledges the limitation, (b) assesses its potential magnitude (see below) and (c) emphasizes that spatially resolved sampling across the Basin would be valuable in future studies. We propose that future efforts sample a transect from the outer harbour, and/or edges of the Bedford Basin to the central Compass Station in order to fully capture spatial heterogeneity and better assess the contribution of lateral inputs on both 234Th and POC flux. The Bedford Basin is connected to the Halifax harbour & open Atlantic, and therefore one might expect some addition of total 234Th from these waters into the Bedford Basin. Although lateral impacts are not directly measured in this study, we anticipate that they are small for the following reasons:

Modelling efforts (Shan and Sheng, 2012) used particle tracking in the Bedford Basin and Halifax Harbour to demonstrate that  $\sim 80\%$  of the particles that originate in the basin stay in the basin. The model implies that the observations in the central basin predominantly reflect vertical production and settling. Lateral exchanges occur only episodically during storms and strong tidal phases, but do not constitute a persistent flux pathway. Additionally, Shan et al. (2011) find that the sill between the Bedford Basin and the outer harbour largely prevents mixing below 20 – 30 m water depth. We performed a simple first-order estimate of lateral advection using observed basin current velocities (surface ~ 0.2 cm s-1, subsurface < 1cm s-1; Shan and Sheng, 2012; Burt et al., 2013), a 1 km basin scale, and small cross-basin gradients in 234Th (0.01–0.03 dpm L-1). This analysis suggests that lateral fluxes contribute typically < 10%, and maximally 20%, of the vertical flux depending on layer and assumed gradient. Additionally, based on numerical passive tracer experiments, Shan and Sheng (2012) estimate the e-folding flushing time of 40 - 90 days in the upper and entire Bedford Basin respectively. In our study, we find the residence time of particulate  $^{234}$ Th to be  $18 \pm 7$  days, which is substantially shorter than, or at best comparable to, the advective flushing timescale. Thus, while episodic lateral exchanges may occur during storms or intrusions, multiple lines of evidence suggest that vertical production and settling dominate 234Th-POC dynamics in the Bedford Basin. In our revised text, we explain in more detail the points. The revised text now reads:

(468) "In this study, we assume that  $\partial^{234}Th/\partial t$  is equal to 0 and the system is in steady state. The SS model assumes that i)  $^{234}Th$  activities and removal rates are near constant with respect to  $^{234}Th$  decay, ii) the contributions of vertical and horizontal advection of  $^{234}Th$  are minor, and iii) scavenging is irreversible. Physical studies of the Bedford Basin have generally shown that 1D assumptions hold for most of the year (Burt et al.,

2013), and modelling efforts have demonstrated that  $\sim$  80% of particles that originate in the basin stay in the basin (Shan and Sheng, 2012). The circulation pattern of Halifax Harbour is characterized as a two-layer, estuarine-type system, with a fresher upper layer flowing seaward, driven by inflows from the Sackville River, and a saltier deep-water return flow (Fader and Miller, 2008). However, with a mean surface outflow of 0.2 cm s-1, this circulation is weakest in the Bedford Basin (Burt et al., 2013). Despite an inflow from the open Atlantic, the presence of a sill largely prevents mixing below 20 m in the basin (Shan et al., 2011). Although our observations come from a single, centrally located station, we acknowledge that lateral heterogeneity and edge effects may influence 234Th and POC dynamics in the Bedford Basin, especially from 0 – 20 m. To assess this, we evaluated a surface lateral 234Th contribution using our 0 – 20 m mean 234Th activity, a 3 km cross-basin scale, and observed basin current velocities (0.2 cm s-1). For a theoretical 50% cross basin difference in 234Th activities, the lateral 234Th flux contribution is ~346 dpm m-2 d-1, which is ~52% of our mean surface cumulative flux, and ~14% of the full water column flux to 65 m. This suggests that even under a strong 50% contrast, the full water column budget changes only modestly. These lateral 234Th flux estimates are intended as a first order, illustrative calculation, and real cross-basin gradients 234Th activities (if any) should be measured for a more robust estimate of lateral contributions in future studies."

The hydrological parameters of the basin show similar patterns for different years and seasonal variations. The mixing and stratification are seasonal and the importance of absence of winter mixing and spring blooms in relation with steady state 1-D Model needs to be further discussed. The limited direct measurements of nepheloid layers and primary production weakened mechanistic interpretation of persistent  $^{234}$ Th deficits. Although 1-D steady state model is assumed in this study, the system shows several seasonal dynamic features, especially intrusion and spring bloom. The authors have included a paragraph of discussion about the NSS contribution for < 3%, but it doesn't justify this datasheet. The authors should describe how the absence of NSS might impact the fluxes especially during different seasons and spring bloom for this study.

**Response:** We thank the reviewer for this important comment. We agree that the Bedford Basin is a seasonally dynamic system with winter mixing, spring blooms, and episodic intrusions, all of which can potentially influence particle fluxes and 234Th distributions. Like many 234Th studies, our study used steady state assumptions due to the logistical constraints associated with field sampling. While we recognize that NSS conditions likely occur during certain periods, the application of the 1-D steady-state model allows for consistency and comparability with previous studies. In a revised version of the manuscript, we have expanded the discussion of NSS as follows:

- In section 4.3, we have expanded our discussion to address how our steady-state fluxes could be impacted by spring bloom and intrusion events.
- We apply a theoretical non-steady state sensitivity to our two spring profiles. Assuming a non-steady state term of 0.003 dpm L-1 d-1, the NSS correction is 7.8 % of our reported SS 234Th flux. To also include an upper boundary spring bloom scenario, we also tested a larger non-steady state term of 0.01 dpm L-1 d-1. The resulting NSS correction is ~ 25.8% of the spring SS estimate we report. We also quantify and outline in 4.3 how these potential NSS contributions change our POC flux estimates.
- We briefly discuss and estimate the possible NSS contribution from intrusion events.

In conclusion, though we did not directly measure NSS impacts, we now report theoretical NSS fluxes in the revised manuscript and expand on their implications in the Bedford Basin. We note that repeat sampling (every 1–4 weeks) remains the preferred way to quantify NSS explicitly when feasible. We suggest that future work in the Bedford Basin should consider following a NSS sampling scheme to better constrain  $^{234}$ Th derived fluxes. Additionally, use of other shorter-lived tracers, such as Bismuth-210 ( $t_{1/2} = 5$  d) can capture even finer resolutions (hours-days). The revised text now reads:

(L486) "NSS impacts (rapid temporal changes in  $^{234}$ Th activities) have been shown to play a role on flux estimates during phytoplankton blooms, and prior studies have shown that NSS terms during blooms can alter  $^{234}$ Th export fluxes often by 10-50% (Buesseler et al., 1992; Ceballos-Romero et al., 2018). To assess how these departures from SS may impact our reported fluxes, we applied theoretical NSS sensitivities to our spring profiles. Assuming

a modest NSS term (0.003 dpm  $L^{-1}$ ), the NSS correction is 7.8% of our reported SS 234Th flux. Under an upper bound scenario with a larger NSS term (0.01 dpm  $L^{-1}$ ), the resulting NSS correction increases to ~25.8% of the spring SS estimate. These theoretical corrections show that though minor, NSS effects can become significant during rapid bloom events. Periodic deep vertical mixing, such as during an intrusion event could increase the potential to disrupt condition of the SS model. Indeed, we do see some indication of mixing multiple times throughout the study period (Fig. 2e), however, no major lines of evidence in the 234Th activities are indicated in our study (no major  $RA_{P234}$  signals, or significant difference in POC:234Th), and despite the intrusion in January 2024, the 234Th activities are similar between January 2024 (intrusion) and January 2022 (no intrusion). If feasible, future studies in the basin should sample on a finer temporal resolution (1 – 4 weeks) to better understand NSS contributions in the basin"

I appreciate the authors for nice graphs and figures; however, the authors should include the 234Th profile data along with errors even if it's in supplementary data.

**Response:** The 234Th profile data is included in Fig. 3. The error bars are small and were possibly missed by the reviewer.

Fig 6 a. There is no need to include the R2 value as this is cumulative fluxes and R2 values doesn't have any meaningful essence.

Response: Change made.

Overall, the manuscript is valuable, relevant and methodologically sound. Incorporating these suggestions, addressing spatial heterogeneity would provide clarity and enhance the impact of this manuscript.

**Works cited:**

- Burt, W. J., Thomas, H., Fennel, K., & Horne, E. (2013). Sediment-water column fluxes of carbon, oxygen and nutrients in Bedford Basin, Nova Scotia, inferred from 224 Ra measurements. Biogeosciences, 10(1), 53–66. https://doi.org/10.5194/bg-10-53-2013
- Shan, S., & Sheng, J. (2012). Examination of circulation, flushing time and dispersion in Halifax Harbour of Nova Scotia. Water Quality Research Journal, 47(3–4), 353–374. https://doi.org/10.2166/wqrjc.2012.041
- Shan, S., Sheng, J., Thompson, K. R., & Greenberg, D. A. (2011). Simulating the three-dimensional circulation and hydrography of Halifax Harbour using a multi-nested coastal ocean circulation model. Ocean Dynamics, 61(7), 951–976. https://doi.org/10.1007/s10236-011